# Conditional Image Synthesis with Diffusion Models: A Survey

**Zheyuan Zhan**                                                    *zhanzheyuan@zju.edu.cn*
*State Key Laboratory of Blockchain and Data Security, Zhejiang University*
*College of Computer Science, Zhejiang University*

**Defang Chen**[†]                                                 *defangch@buffalo.com*
*University at Buffalo, State University of New York*

**Jian-Ping Mei**                                                  *jpmei@zjut.edu.cn*
*College of Computer Science, Zhejiang University of Technology*

**Zhenghe Zhao**                                                   *zhaozhenghe@zju.edu.cn*
*College of Computer Science, Zhejiang University*

**Jiawei Chen**                                                    *sleepyhunt@zju.edu.cn*
*State Key Laboratory of Blockchain and Data Security, Zhejiang University*
*Hangzhou High-Tech Zone (Binjiang) Institute of Blockchain and Data Security*

**Chun Chen**                                                      *chenc@zju.edu.cn*
*College of Computer Science, Zhejiang University*

**Siwei Lyu**                                                      *siweilyu@buffalo.com*
*University at Buffalo, State University of New York*

**Can Wang**                                                       *wcan@zju.edu.cn*
*State Key Laboratory of Blockchain and Data Security, Zhejiang University*
*Hangzhou High-Tech Zone (Binjiang) Institute of Blockchain and Data Security*

**Reviewed on OpenReview:** *https://openreview.net/forum?id=ewwNKwh6SK*

## Abstract

Conditional image synthesis based on user-specified requirements is a key component in creating complex visual content. In recent years, diffusion-based generative modeling has become a highly effective way for conditional image synthesis, leading to exponential growth in the literature. However, the complexity of diffusion-based modeling, the wide range of image synthesis tasks, and the diversity of conditioning mechanisms present significant challenges for researchers to keep up with rapid developments and to understand the core concepts on this topic. In this survey, we categorize existing works based on how conditions are integrated into the two fundamental components of diffusion-based modeling, *i.e.*, the denoising network and the sampling process. We specifically highlight the underlying principles, advantages, and potential challenges of various conditioning approaches during the training, re-purposing, and specialization stages to construct a desired denoising network. We also summarize six mainstream conditioning mechanisms in the sampling process. All discussions are centered around popular applications. Finally, we pinpoint several critical yet still unsolved problems and suggest some possible solutions for future research. Our reviewed works are itemized at `https://github.com/zju-pi/Awesome-Conditional-Diffusion-Models`.

---

[†]Corresponding Author.

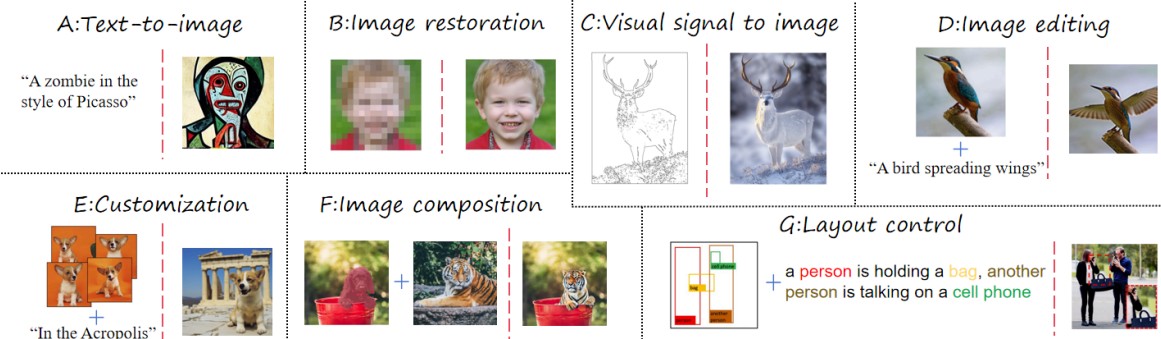

Figure 1: Seven representative conditional image synthesis tasks with their corresponding inputs and outputs. Figures are cited from the following papers: (A) Stable Diffusion (Rombach et al., 2022); (B) SR3 (Saharia et al., 2022c); (C) ControlNet (Zhang et al., 2023b); (D) Imagic (Kawar et al., 2023); (E) DreamBooth (Ruiz et al., 2023); (F) PbE (Yang et al., 2023a); (G) InteractDiffusion (Hoe et al., 2023).

## 1 Introduction

Image synthesis is an essential task in generative artificial intelligence. It is particularly useful when user-provided conditional inputs guide the generation process, enabling precise control to meet diverse needs. Early works have achieved significant breakthroughs in various conditional image synthesis tasks, such as text-to-image generation (Reed et al., 2016; Zhang et al., 2017; Ding et al., 2021; Ramesh et al., 2021), image restoration (Ledig et al., 2017; Wang et al., 2021; Maaløe et al., 2019; Lee et al., 2022), and image editing (Brock et al., 2017; Ling et al., 2021; Abdal et al., 2020). However, early deep learning-based generative models, such as generative adversarial networks (GANs) (Goodfellow et al., 2014; Mirza & Osindero, 2014), variational auto-encoders (VAEs) (Kingma & Welling, 2014; Sohn et al., 2015), and auto-regressive models (ARMs) (Van Den Oord et al., 2016; Van den Oord et al., 2016) face inherent limitations. GANs are susceptible to mode collapse and unstable training (Goodfellow et al., 2014); VAEs often produce blurry images (Kingma & Welling, 2014); and ARMs suffer from sequential error accumulation and significant time delays (Van Den Oord et al., 2016).

In recent years, diffusion models (DMs) have emerged as the state-of-the-art framework for image generation due to their strong generative capabilities and versatility (Sohl-Dickstein et al., 2015; Ho et al., 2020; Song et al., 2021b; Karras et al., 2022; Chen et al., 2024a). In DMs, images are generally synthesized from Gaussian noise through iterative denoising steps guided by the predictions of a denoising network. In practice, DMs achieve remarkable performance in diverse generative tasks, characterized by stable training process, diverse outputs, and exceptional sample quality. Furthermore, compared to one-step generative models, the distinctive multi-step sampling process offers DMs a unique advantage in facilitating conditional integration. These benefits have made DMs a preferred tool for conditional image synthesis, leading to a rapid growth in *Diffusion-based Conditional Image Synthesis* (DCIS) research over the past few years (Rombach et al., 2022; Saharia et al., 2022b; Lu et al., 2023; Choi et al., 2021; Saharia et al., 2022c; Kawar et al., 2023; Hertz et al., 2023; Zhang et al., 2023e; Gal et al., 2023a; Zhang et al., 2023b; Wang et al., 2024b). Fig. 1 illustrates seven popular DCIS tasks with different modalities of conditional inputs.

With the rapid growth of research in this area, coupled with the wide variability in model architectures, training paradigms, and sampling strategies, as well as the broad scope of potential conditional synthesis tasks, it has become increasingly challenging for newcomers to develop a comprehensive understanding of the landscape of DCIS. This underscores the need for a systematic survey that offers a coherent and structured synthesis of current advances in this rapidly evolving field.

Several surveys have focused on the application of diffusion models in specific conditional image synthesis tasks, such as image restoration (Li et al., 2023g; Daras et al., 2024), text-to-image (Zhang et al., 2023a), and image editing (Huang et al., 2024b), or have categorized diffusion-based works in the field of computer vision according to their target conditional synthesis tasks (Croitoru et al., 2023; Po et al., 2023). While

these task-oriented surveys provide valuable insights into approaches tailored for different tasks, they do not explore the commonalities in model frameworks across different conditional synthesis tasks, particularly in terms of model architectures and conditioning mechanisms.

Two recent surveys (Shuai et al., 2024; Cao et al., 2024) provide an overview of DM-based works across a broad range of conditional image synthesis tasks. However, their scope remains limited, as they primarily focus on DCIS methods built on text-to-image (T2I) backbones, neglecting earlier works that integrate conditional inputs into unconditional denoising networks or train task-specific conditional denoising networks from scratch. These earlier efforts are foundational to the current advancements in DCIS and are still widely applied in low-level tasks such as image restoration.

In contrast, this survey aims to provide a comprehensive and structured framework that covers a wide range of contemporary DCIS works. We present a taxonomy based on the mainstream techniques for condition integration, offering a clear and systematic breakdown of the key components and design choices involved in constructing a DCIS framework. Specifically, we review and categorize existing DCIS methods by examining how conditions are integrated into the two fundamental components of diffusion modeling: the *denoising network* and the *sampling process*. For the denoising network, we delineate the process of establishing a conditional denoising network into three stages. For the sampling process, we categorize six mainstream in-sampling conditioning mechanisms, detailing how control signals are integrated into various components of the sampling process. Our objective is to provide readers with a high-level and accessible overview of existing DCIS works across diverse tasks, equipping them with the knowledge to design conditional synthesis frameworks for their own applications, including novel tasks that have yet to be explored. In practice, as image synthesis is a fundamental task in computer vision, many more complex visual computing and synthesis tasks are built upon its extensions. Therefore, the methods for image synthesis introduced in this paper can be readily extended to more complex visual tasks, such as video synthesis (Wang et al., 2023c; Esser et al., 2023), 3D scene generation (Haque et al., 2023; Höllein et al., 2023), motion generation (Karunratanakul et al., 2023; Kulkarni et al., 2024).

The remainder of this survey is organized as follows: We first introduce the background of diffusion models and conditional image synthesis in Sec. 2. Next, we summarize methods for condition integration within the denoising network in Sec. 3, and for the sampling process in Sec. 4. Finally, we outlines potential future directions in Sec. 5.

## 2 Backgrounds

Diffusion-based generative modeling adopts a forward diffusion process of gradually adding noise into clean data and learns a denoising network to predict the added noise. In the sampling process, the data is synthesized by reversing the forward process from Gaussian noise based on the prediction of a denoising network. Currently, a branch of conditional synthesis research (Esser et al., 2024; Tewel et al., 2024; Wang et al., 2024a; Rout et al., 2024a) leverages the flow matching framework (Lipman et al., 2023; Liu et al., 2023c; Heitz et al., 2023) to model the mapping from a prior distribution to the real data distribution. In practice, most of these works employ a special case of flow matching, where the prior distribution used in flow matching corresponds to a Gaussian distribution, resulting in the same training and sampling algorithm as the original diffusion framework in practice (Gao et al., 2024). Therefore, we primarily focus on the generative process based on the original diffusion framework, unless otherwise specified. We first introduce the core concepts of discrete-time and continuous-time diffusion modeling in Sec. 2.1. Then, we discuss the model architecture in Sec. 2.2 and highlight representative DCIS tasks in Sec. 2.3. Finally, in Sec. 2.4, we introduced the classic condition strengthening approaches widely employed across various DCIS works.

### 2.1 The Formulation of Diffusion Modeling

### 2.1.1 Discrete-Time Formulation

The discrete-time diffusion model was initially proposed in (Sohl-Dickstein et al., 2015). It constructs a forward Markov chain to transform clean data into noise by progressively adding small amounts of Gaussian noise so that a parameterized denoising network can be learned to predict the added noise in each forward

step. Once the denoising network is trained, images can be generated from Gaussian noise by reversing the diffusion process. This idea gained popularity through an important follow-up work known as denoising diffusion probabilistic models (DDPMs) (Ho et al., 2020). This work led to a substantial improvement in the quality of synthesized images in increased resolutions, from $32 \times 32$ (Sohl-Dickstein et al., 2015) to $256 \times 256$, sparking a rapidly growth of interest in diffusion models. Next, we adopt the notation from DDPM (Ho et al., 2020), which is widely employed in the literature to describe discrete-time diffusion models (Song et al., 2021a; Rombach et al., 2022; Kawar et al., 2023).

The forward Markov chain is parameterized based on a pre-defined schedule $\beta_1, \ldots, \beta_T$, where $\beta_t$ is the noise variance in each step and the total number of steps $T$ is usually large, *e.g.*, 1,000. Given the clean data sampled from the training dataset $\mathbf{x}_0 \sim p_{\text{data}}(\mathbf{x})$, the transition kernel is $q(\mathbf{x}_t \mid \mathbf{x}_{t-1}) = \mathcal{N}(\mathbf{x}_t; \sqrt{1 - \beta_t}\mathbf{x}_{t-1}, \beta_t\mathbf{I})$, or, $q(\mathbf{x}_t \mid \mathbf{x}_0) = \mathcal{N}(\mathbf{x}_t; \sqrt{\bar{\alpha}_t}\mathbf{x}_0, (1 - \bar{\alpha}_t)I)$, where $\mathbf{x}_1, \ldots, \mathbf{x}_T$ are latent variables, $\alpha_t = 1 - \beta_t$, $\bar{\alpha}_t = \prod_{i=1}^t \alpha_i$, and $\bar{\alpha}_T \to 0$. By progressively adding Gaussian noise to the clean data, this Markov chain transforms the data distribution to an approximate normal distribution, *i.e.*, $\int q(\mathbf{x}_T|\mathbf{x}_0)p_{\text{data}}(\mathbf{x}_0)\mathrm{d}\mathbf{x}_0 \approx \mathcal{N}(0, \mathbf{I})$.

In the training phase, DDPM (Ho et al., 2020) learns a denoising network with parameter $\boldsymbol{\theta}$ by minimizing the KL divergence between the transition kernel $p_{\boldsymbol{\theta}}(\mathbf{x}_{t-1}|\mathbf{x}_t)$ and the posterior distribution $q(\mathbf{x}_{t-1} \mid \mathbf{x}_t, \mathbf{x}_0)$. In practice, DDPM (Ho et al., 2020) is trained on the following re-parameterized loss function to improve the training stability and sample quality:

$$\mathbb{E}_{t,\mathbf{x}_0,\boldsymbol{\epsilon}}\left[\left\|\boldsymbol{\epsilon} - \boldsymbol{\epsilon}_{\boldsymbol{\theta}}\left(\sqrt{\bar{\alpha}_t}\mathbf{x}_0 + \sqrt{1 - \bar{\alpha}_t}\boldsymbol{\epsilon}, t\right)\right\|_2^2\right], \tag{1}$$

where $\boldsymbol{\epsilon}_{\boldsymbol{\theta}}(\mathbf{x}_t, t)$ is a noise-prediction network to estimate the added noise $\boldsymbol{\epsilon} = \frac{\mathbf{x}_t - \sqrt{\bar{\alpha}_t}\mathbf{x}_0}{\sqrt{1-\bar{\alpha}_t}}$ in each step. For the conditional generation that performs denoising steps conditioned on control signals $\mathbf{c}$, the conditional denoising network $\boldsymbol{\epsilon}_{\boldsymbol{\theta}}(\mathbf{x}_t, t, \mathbf{c})$ can be trained on a loss function similar to Eq. 1:

$$\mathbb{E}_{t,\mathbf{c},\mathbf{x}_0 \sim p(\mathbf{x}_0|\mathbf{c}),\boldsymbol{\epsilon}}\left[\left\|\boldsymbol{\epsilon} - \boldsymbol{\epsilon}_{\boldsymbol{\theta}}\left(\sqrt{\bar{\alpha}_t}\mathbf{x}_0 + \sqrt{1 - \bar{\alpha}_t}\boldsymbol{\epsilon}, t, \mathbf{c}\right)\right\|_2^2\right]. \tag{2}$$

In the sampling process, DDPM gradually generates clean data from Gaussian noise by computing the reverse transition kernel $p_{\boldsymbol{\theta}}$ with the learned network $\boldsymbol{\epsilon}_{\boldsymbol{\theta}}$, i.e.,

$$\mathbf{x}_{t-1} = \frac{1}{\sqrt{\alpha_t}}\left(\mathbf{x}_t - \frac{1 - \alpha_t}{\sqrt{1 - \bar{\alpha}_t}}\boldsymbol{\epsilon}_{\boldsymbol{\theta}}\right) + \frac{1 - \bar{\alpha}_{t-1}}{1 - \bar{\alpha}_t}\beta_t\boldsymbol{\epsilon}_t, \tag{3}$$

where $\boldsymbol{\epsilon}_t \sim \mathcal{N}(0, \boldsymbol{I})$ is the standard Gaussian noise independent of $\mathbf{x}_t$. The following work DDIM (Song et al., 2021a) proposed a family of sampling processes sharing the same marginal distribution $p(\mathbf{x}_t)$ with the above sampling process, which are written as

$$\mathbf{x}_{t-1} = \sqrt{\bar{\alpha}_{t-1}} \cdot \boldsymbol{f}_{\boldsymbol{\theta}}(\mathbf{x}_t) + \sqrt{1 - \bar{\alpha}_{t-1} - \sigma_t^2} \cdot \boldsymbol{\epsilon}_{\boldsymbol{\theta}} + \sigma_t\boldsymbol{\epsilon}_t, \tag{4}$$

where $\boldsymbol{f}_{\boldsymbol{\theta}}(\mathbf{x}_t) = \frac{\mathbf{x}_t - \sqrt{1-\bar{\alpha}_t}\boldsymbol{\epsilon}_{\boldsymbol{\theta}}}{\sqrt{\bar{\alpha}_t}}$ denotes the predicted $\mathbf{x}_0$ at time step $t$. For simplicity, we will refer to $\boldsymbol{f}_{\boldsymbol{\theta}}(\mathbf{x}_t)$ as the intermediate denoising output $\mathbf{x}_{0|t}$ hereafter. Each choice of $\sigma_t$ represents a specific sampling process in DDIM (Song et al., 2021a). It is identical to the DDPM generative process in Eq. 3 when $\sigma_t = \sqrt{(1 - \bar{\alpha}_{t-1})/(1 - \bar{\alpha}_t)} \cdot \sqrt{1 - \bar{\alpha}_t/\bar{\alpha}_{t-1}}$ and becomes a deterministic process when $\sigma_t = 0$.

### 2.1.2 Continuous-Time Formulation

Song et al. (2021b) proposed to formulate a diffusion process $\{\mathbf{x}_t \sim p_t(\mathbf{x})\}_{t=0}^T$ with the continuous time variable $t \in [0, T]$ as the solution of an Itô stochastic differential equation (SDE) $d\mathbf{x} = \mathbf{f}(\mathbf{x}, t)dt + g(t)d\mathbf{w}_t$, where $\mathbf{w}_t$ denotes the standard Wiener process, and $\mathbf{f}(\mathbf{x}, t)$ and $g(t)$ are drift and diffusion coefficients, respectively (Oksendal, 2013; Chen et al., 2024a). This diffusion process smoothly transforms a data distribution into an approximate noise distribution $p_T$, and the forward process of DDPM (Ho et al., 2020) can be regarded as a specific discretization of it. There exists a probability flow ordinary differential equation (PF-ODE) $d\mathbf{x} = \left[\mathbf{f}(\mathbf{x}, t) - \frac{1}{2}g(t)^2\nabla_{\mathbf{x}} \log p_t(\mathbf{x})\right]dt$, sharing the same marginal distribution with the reverse

SDE $d\mathbf{x} = \left[\mathbf{f}(\mathbf{x}, t) - g(t)^2 \nabla_{\mathbf{x}} \log p_t(\mathbf{x})\right] dt + g(t) d\hat{\mathbf{w}}$ (Song et al., 2021b; Karras et al., 2022; Zhang & Chen, 2023; Chen et al., 2024a). We can learn a time-dependent score-based denoising network $\mathbf{s}_{\boldsymbol{\theta}}(\mathbf{x}_t, t)$ to estimate the score function $\nabla_{\mathbf{x}_t} \log p(\mathbf{x}_t)$ with a sum of denoising score matching (Vincent, 2011; Lyu, 2009) objectives weighted by $\lambda(t)$:

$$\mathbb{E}_t \left[\lambda(t) \mathbb{E}_{\mathbf{x}_0, \mathbf{x}_t} \left[\|\mathbf{s}_{\boldsymbol{\theta}}(\mathbf{x}_t, t) - \nabla_{\mathbf{x}} \log p(\mathbf{x}_t \mid \mathbf{x}_0)\|_2^2\right]\right]. \tag{5}$$

When the score-based denoising network $\mathbf{s}_{\boldsymbol{\theta}}(\mathbf{x}_t, t)$ is trained, we can employ general-purpose numerical methods such as Euler-Maruyama and Runge-Kutta methods to solve the reverse SDE or PF-ODE and recover clean data $\mathbf{x}_0$ from $\mathbf{x}_T$.

In practice, learning a score-based denoising network $\mathbf{s}_{\boldsymbol{\theta}}$ or a noise-prediction network $\boldsymbol{\epsilon}_{\boldsymbol{\theta}}$ are essentially equivalent(It can be proven that $\boldsymbol{\epsilon}_{\boldsymbol{\theta}}$ approximates a scaled score function $-\sqrt{1 - \bar{\alpha}_t} \nabla_{\mathbf{x}_t} \log p(\mathbf{x}_t)$). The DDPM sampling process described in Eq. 3 can be regarded as a first-order numerical solution for the reverse SDE. Therefore, in the following sections, unless otherwise specified, we will use the notation $\boldsymbol{\epsilon}_{\boldsymbol{\theta}}$ to represent the denoising network.

## 2.2 Architecture of the Denoising Network

Pioneering works adopt U-Net (Ronneberger et al., 2015) architectures as the backbone of denoising networks (Ho et al., 2020; Song et al., 2021a; Song & Ermon, 2019; 2020). As illustrated in Fig.2, the denoising network employed in DDPM (Ho et al., 2020) follows the U-shaped structure of downsampling and upsampling blocks in the basic U-Net. At each resolution level, features from the downsampling blocks are directly passed to the corresponding upsampling blocks through skip connections, which helps retain high-resolution local information and prevent the loss of details during the upsampling process. To help the denoising network to better capture visual features and the pixel correlations, the denoising network also replaces the simple convolution layers in the original U-Net with residual convolution layers (Zagoruyko, 2016) and self-attention layers.

The U-Net architecture is particularly well-suited for diffusion models due to its ability to perform superior feature extraction, contextual understanding, precise segmentation, and dimensionality preservation. These attributes enable it to accurately model complex data distributions and generate high-quality images. Building on this foundation, many followed-up works have developed more advanced U-Net-based denoising networks by incorporating multi-head attention (Song et al., 2021b; Dhariwal & Nichol, 2021; Nichol & Dhariwal, 2021), normalization (Ho et al., 2020; Dhariwal & Nichol, 2021; Nichol & Dhariwal, 2021), and cross-attention layers (Rombach et al., 2022; Saharia et al., 2022b). Transformers, known for their scalability, robustness, and efficiency, have also emerged as a popular

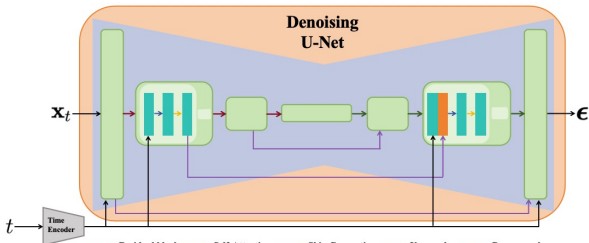

Figure 2: An illustration of the DDPM denoising network (Ho et al., 2020), which predicts the noise $\boldsymbol{\epsilon}$ based on the given latent variable $\mathbf{x}_t$ and time step $t$. The timestep $t$ is firstly converted to high-dimensional representations via a time encoder(e.g.,sinusoidal embeddings and MLPs) and subsequently added to intermediate feature maps.

choice for the model architecture in a variety of computer vision tasks. Researchers have attempted to employ transformer architectures as the backbone of denoising networks in various conditional synthesis tasks (Yang et al., 2022b; Tang et al., 2022; Gu et al., 2022; Li et al., 2023d). However, these initial efforts have yet to rival the dominance of U-Net architectures in diffusion models. Notably, the recent groundbreaking Diffusion Transformers (DiTs) (Peebles & Xie, 2023), built on the Vision Transformer (ViT) (Dosovitskiy et al., 2020) architecture, first convert spatial input into a sequence of tokens and then process them through a series of transformer blocks. The timestep and class label in DiTs are integrated via adaptive layer normalization. In practice, DiTs achieve state-of-the-art sample quality, surpassing all previous diffusion models at comparable computational costs.

Despite the impressive generative performance of transformer-based model architectures, most DCIS works still adopt U-Net as the model structure. Therefore, in the following sections, unless explicitly stated otherwise, we assume that the denoising network follows a U-Net structure.

### 2.3 Conditional Image Synthesis Tasks

A conditional image synthesis task $\mathcal{T}$ generates target image $\mathbf{x}$ by sampling from a conditional distribution:

$$\mathbf{x} \sim p_{\mathcal{T}}(\mathbf{x}|\mathbf{c}), \ \mathbf{c} \in \mathcal{D}_{\mathcal{T}}, \tag{6}$$

where $\mathcal{D}_{\mathcal{T}}$ is the domain of conditional inputs $\mathbf{c}$, and $p_{\mathcal{T}}$ is the conditional distribution defined by the task $\mathcal{T}$. Based on the form of conditional inputs $\mathbf{c}$ and the correlation between the conditional inputs and the target image formulated as conditional distribution $p_{\mathcal{T}}(\mathbf{x}|\mathbf{c})$, we classify representative conditional image synthesis tasks into seven categories as shown in Fig. 1: (a) *Text-to-image* synthesizes images in accordance with text prompts, (b) *Image restoration* recovers clean images from their degraded counterparts, (c) *visual signal to image* converts given visual signals such as sketch, depth and human pose into corresponding images, (d) *Image editing* edits the given source images with provided semantic, structure or style information, (e) *Customization* creates different editing renditions for personal objects specified by given images, (f) *Image composition* composes the objects and the background specified in different images into a single image, and (g) *Layout control* controls the layout grounding of synthesized images with provided spatial information of foreground objects and background. Further qualitative comparisons between classic DCIS works for text-to-image, image restoration, visual signal to image, image editing and customization are provided in Fig. 11, 12, 13, 14, 15 in the Appendix. For image composition and layout control, due to the varying formats of conditional inputs across different works, a direct comparison is not feasible. Therefore, we present only representative outputs from popular works in Fig. 16, 17. Besides, we have sorted out the associations between the conditional synthesis tasks and the representative conditioning mechanisms in Tab. 1.

### 2.4 Condition Strengthening in the Sampling Process

Currently, in order to strengthen the influence of the given conditional inputs $\mathbf{c}$ in the synthesized image, numerous DCIS works attempt to sample from the conditional strengthened distribution $p(\mathbf{x}|\mathbf{c})p(\mathbf{c}|\mathbf{x})^w$ rather than the original conditional distribution $p(\mathbf{x}|\mathbf{c})$. In this formula, the parameter $w$ controls the strength of conditional inputs $\mathbf{c}$, which leads to a trade-off between sample quality and diversity. In practice, setting a large scaling factor $w$ can significantly enhance the sample quality and the consistency with the conditional inputs $\mathbf{c}$ at the cost of sample diversity (Dhariwal & Nichol, 2021; Ho & Salimans, 2022).

Classifier Guidance (Dhariwal & Nichol, 2021) trains an auxiliary classifier $p_{\phi}(\mathbf{c} \mid \mathbf{x}_t)$ to approximate the likelihood term $p(\mathbf{c} \mid \mathbf{x}_t)$ in label-conditioned image synthesis. However, training an accurate classifier in most of the conditional synthesis tasks is challenging. Classifier-free guidance (Ho & Salimans, 2022) paves a training-free pathway to approximate $p(\mathbf{c} \mid \mathbf{x}_t) \propto p(\mathbf{x}_t \mid \mathbf{c})/p(\mathbf{x}_t)$ with the access to conditional noise prediction $\boldsymbol{\epsilon}_{\boldsymbol{\theta}}(\mathbf{x}_t, \mathbf{c}) = -\sqrt{1 - \bar{\alpha}_t}\nabla_{\mathbf{x}_t}\log p(\mathbf{x}_t|\mathbf{c})$ and the unconditional noise prediction $\boldsymbol{\epsilon}_{\boldsymbol{\theta}}(\mathbf{x}_t) = -\sqrt{1 - \bar{\alpha}_t}\nabla_{\mathbf{x}_t}\log p(\mathbf{x}_t)$. The proxy noise prediction $\tilde{\boldsymbol{\epsilon}}_{\boldsymbol{\theta}}$ can be expressed as:

$$\tilde{\boldsymbol{\epsilon}}_{\boldsymbol{\theta}}(\mathbf{x}_t, \mathbf{c}) = (1 + w)\boldsymbol{\epsilon}_{\boldsymbol{\theta}}(\mathbf{x}_t, \mathbf{c}) - w\boldsymbol{\epsilon}_{\boldsymbol{\theta}}(\mathbf{x}_t). \tag{7}$$

Due to its convenience and effectiveness, classifier-free guidance has become the mainstream condition strengthening approach in various DCIS works. To alleviate the potential negative impact of large guidance scales on sample diversity, subsequent works (Sadat et al., 2024; Kynkäänniemi et al., 2024) propose dynamic classifier-free guidance, in which the guidance scaling factor is reduced during the denoising process with high noise levels.

In practice, classifier guidance and classifier-free guidance can also be employed as conditioning mechanisms to inject conditional inputs into the diffusion-based image synthesis framework. Therefore, we incorporate them into our DCIS framework and provide a more detailed discussion in Sec.4.5 and Sec.4.3.

Table 1: Stack of conditioning mechanisms applied to denoising network and sampling process, for mainstream conditional synthesis tasks Conditioning encoder indicates the module to convert conditional inputs into task-related feature embedding, where * indicates that the encoder is determined by the specific restoration task. ♠, ♡, ♣, ♢ denote the four condition injection methods employed in re-purposing stage as described in Sec. 3.2.2. Due to page width limitations, we have placed the DCIS works performing condition integration via the presented stacks of conditioning mechanisms in the row identified by corresponding serial numbers in Tab. 4 in Appendix.

| Stack of conditioning mechanisms for denoising network | | | | | | |
|---|---|---|---|---|---|---|
| Task | Training (backbone) | Conditional encoder | Condition Injection | Backbone fine-tuning | Specialization | Serial Number |
| Text-to-image | ✓ | CLIP, BERT, LLMs | ♡ | ✗ | ✗ | DN1 |
| Image restoration | ✓ | Non. | ♠ | ✗ | ✗ | DN2 |
| | ✓ | * | ♠, ♡ | ✗ | ✗ | DN3 |
| Image editing | ✗ (T2I DM) | LLMs-based | ♡ | ✓ | ✗ | DN4 |
| | ✗ (T2I DM) | Non. | ♠ | ✓ | ✗ | DN5 |
| | ✗ (T2I DM) | Non./BLIP | ♡ | ✗ | ✓ | DN6 |
| Customization | ✗ (T2I DM) | ViT (CLIP)-based | ♡, ♢ | ✗ | Optional | DN7 |
| | ✗ (T2I DM) | Non. | ♡ | ✗ | ✓ | DN8 |
| Visual to image | ✗ (T2I DM) | Convolution-based | ♣ | ✗ | ✗ | DN9 |
| | ✗ (T2I DM) | ViT-based | ♡ | ✗ | ✗ | DN10 |
| Image composition | ✗ (T2I DM) | Convolution-based | ♡ | ✓ | ✗ | DN11 |
| | ✗ (T2I DM) | ViT (CLIP)-based | ♡, ♢ | ✓ | ✗ | DN12 |
| Layout control | ✗ (T2I DM) | ViT (CLIP)-based | ♢ | ✗ | ✗ | DN13 |
| Stack of conditioning mechanisms for sampling process | | | | | | |
| Task | Backbone model | | Conditioning mechanism | | Serial Number | |
| Text-to-image | Uncond DM | | Guidance | | SP1 | |
| Image restoration | Conditional restoration DM | | Revising Diffusion Process | | SP2 | |
| | Uncond DM | | Revising Diffusion Process | | SP3 | |
| | Uncond DM | | Guidance | | SP4 | |
| | Uncond DM | | Conditional Correction | | SP5 | |
| Image editing | Uncond DM / T2I DM | | Inversion | | SP6 | |
| | T2I DM | | Inversion, Conditional Correction | | SP7 | |
| | T2I DM | | Inversion, Attention Manipulation | | SP8 | |
| | T2I DM | | Inversion, Attention Manipulation, Guidance | | SP9 | |
| Visual to image | T2I DM | | Guidance | | SP10 | |
| Image composition | Uncond DM | | Noise Blending | | SP11 | |
| Layout control | T2I DM | | Attention Manipulation | | SP12 | |
| | T2I DM | | Attention Manipulation, Guidance | | SP13 | |
| General purpose | Unspecified | | Noise Composition | | SP14 | |
| | Unspecified | | Classifier-free Guidance | | SP15 | |
| | Unspecified | | Universal Guidance Framework | | SP16 | |

# 3 Condition integration in denoising networks

The denoising network is the crucial component in the diffusion model(DM)-based synthesis framework, which estimates the noise added in each forward step to reverse the Gaussian distribution back into the data

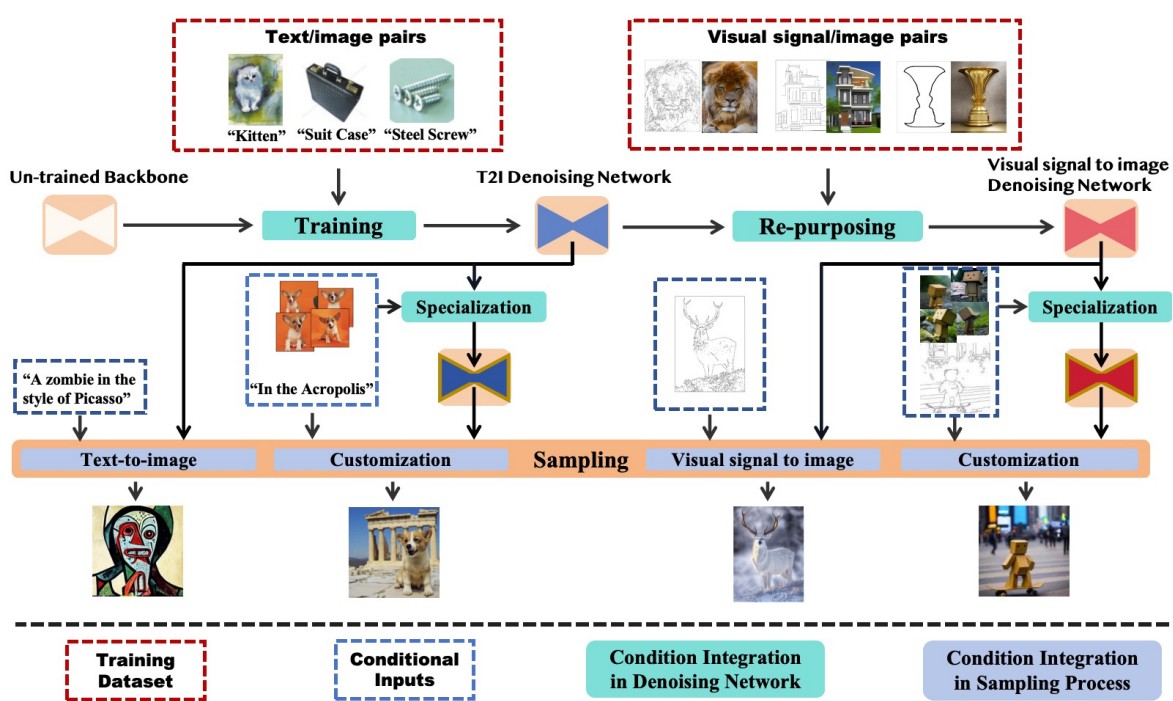

Figure 3: An example of the workflow to build denoising network via training, re-purposing and specialization stages for target conditional synthesis tasks. In this framework, a text-to-image(T2I) denoising network is firstly obtained via supervised learning on text/image pairs in *training stage*. Subsequently, this T2I denoising network is fine-tuned on visual signal/image pairs for visual signal to image task in *re-purposing stage*. Next, both T2I and visual signal to image denoising networks can be further fine-tuned on given images in *specialization stage* to perform customization on the user-specified personal object. Figures are cited from (Rombach et al., 2022; Zhang et al., 2023b; Ruiz et al., 2023; Li et al., 2023a).

Table 2: A Comparison of the characteristics of the three stages to perform condition integration in denoising network. The "Training Cost" column reflects the computational cost involved in establishing a denoising network for the target task, while the "Inference Cost" column represents the computational cost required to customize the denoising network for user-specified conditional inputs. We further present the guarantees of synthesis quality and the commonly applicable task scope(with capital letters indicating the tasks shown in Fig.1) in this table.

| Comparison of conditioning mechanisms for denoising network | | | | |
|---|---|---|---|---|
| **Stage** | **Training Cost** | **Inference Cost** | **Guarantees** | **Applied scope** |
| **Training** | High | No additional cost | High | A,B |
| **Re-purposing** | Medium | No additional cost | Medium | C,D,E,F,G |
| **Specialization** | No additional cost | A relatively high fine-tuning cost | High | D,E |

distribution. In practice, the most straightforward way to achieve conditional control in DM-based synthesis framework is incorporating the conditional inputs into the denoising network. In this section, we divide the condition integration in denoising network into three stages: (a) *training stage*: training a denoising network on paired conditional input and target image from scratch, (b) *re-purposing stage*: re-purposing a pre-trained denoising network to conditional synthesis scenarios beyond the task it was trained on, (c) *specialization stage*: performing testing-time adjustments on denoising network based on user-specified conditional input.

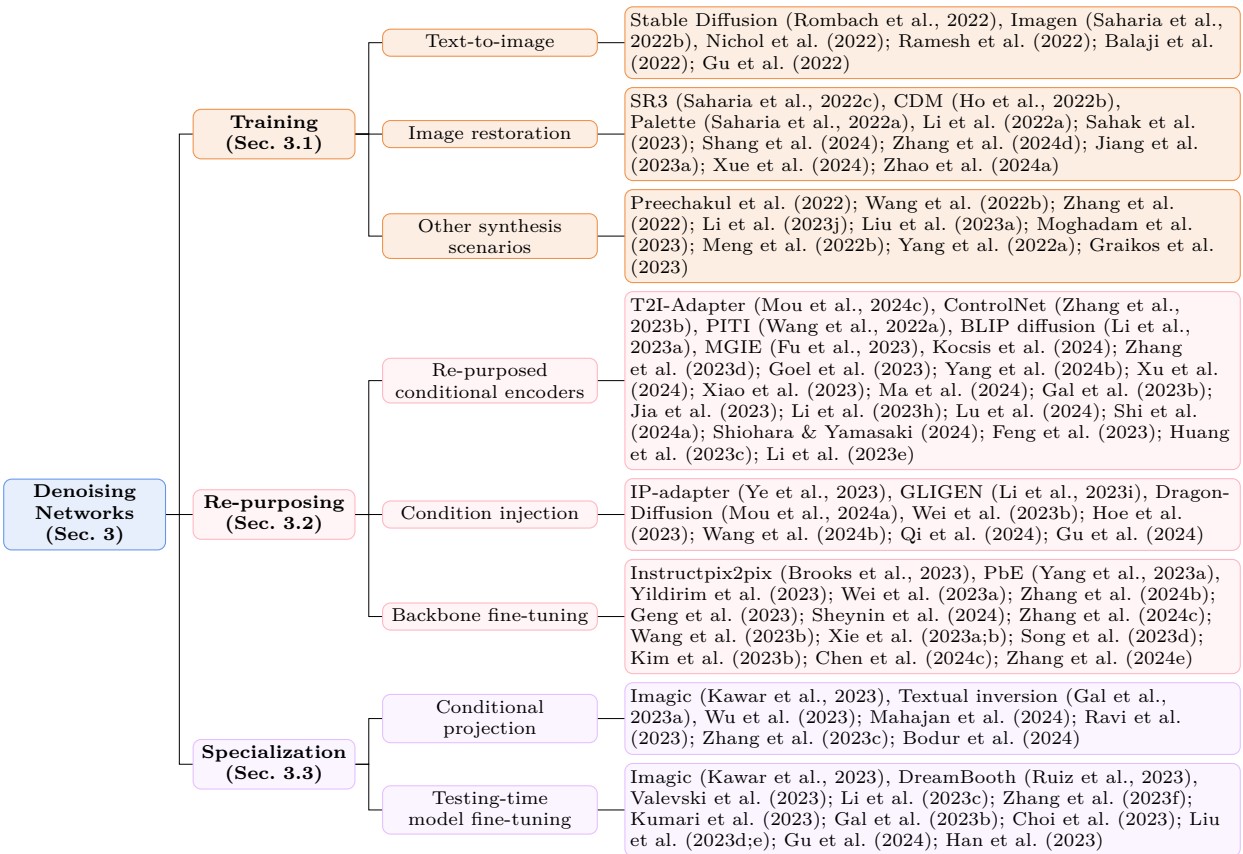

Figure 4: The proposed taxonomy of DCIS works performing condition integration in denoising network.

In practice, the *training stage* is often employed for condition integration in fundamental conditional image synthesis tasks such as image restoration (Saharia et al., 2022c;a; Li et al., 2022a) and text-to-image (Rombach et al., 2022; Ho et al., 2022a; Peebles & Xie, 2023). This stage establishes a reliable relationship between conditional inputs and target images albeit at a high computational cost due to the need for training from scratch. Given the substantial training cost, a branch of works (Zhang et al., 2023b; Li et al., 2023i; Zhang et al., 2023d; Li et al., 2023a) opt to fine-tune a pre-trained text-to-image denoising network to more complicated conditional synthesis tasks via a *re-purposing stage*. This strategy skips the training process and significantly reduces computational cost. However, the relationship between novel conditional inputs and target images re-established during the re-purposing stage is generally less reliable compared to training from scratch. In highly personalized tasks such as customization (Lin et al., 2024a; Gal et al., 2023a) and image editing (Kawar et al., 2023), the task-oriented denoising networks established through the training and re-purposing stages often fail to accurately reproduce fine-grained features from the given conditional inputs. In these cases, the *specialization stage* introduces time-consuming fine-tuning during inference time to align user-specific conditional inputs with the prior knowledge embedded in the denoising network, thereby ensuring detailed consistency between the synthesized image and the provided conditional inputs. We provide a high-level comparison of the pros and cons of performing conditional integration into the denoising network at each stage in Tab. 2.

Fig. 3 provides an examplar workflow to build desired denoising network for conditional synthesis tasks including text-to-image, visual signal to image and customization via these three condition integration stages.

Next, we first review the fundamental conditional DMs modeled in the *training stage* in Sec. 3.1. We then summarize the architecture design choices and conditioning mechanisms for the *re-purposing stage* in Sec. 3.2. Finally, we introduce the works performing condition integration in the *specialization stage* in Sec. 3.3. Fig. 4 illustrates the taxonomy proposed in this section.

### 3.1 Condition Integration in the Training Stage

The most straightforward way to integrate the conditional control signal $\mathbf{c}$ into the denoising network is performing supervised training from scratch with the following loss function:

$$\mathbb{E}_{\mathbf{c},\mathbf{x}\sim p(\mathbf{x}|\mathbf{c}),\boldsymbol{\epsilon},t}\left[\|\boldsymbol{\epsilon} - \boldsymbol{\epsilon}_\theta\left(\mathbf{x}_t,t,\mathbf{c}\right)\|_2^2\right], \tag{8}$$

where $\mathbf{c}$ and $\mathbf{x}$ denote the paired conditional inputs and target image. Thereby, the learned conditional denoising network $\boldsymbol{\epsilon}_{\boldsymbol{\theta}}\left(\mathbf{x}_t,t,\mathbf{c}\right)$ can be employed to sample from $p(\mathbf{x}|\mathbf{c})$.

Next, we introduce the existing conditional denoising networks trained from scratch, focusing on their model architectures and conditioning mechanisms, which are crucial for creating the connection between the conditional inputs and its corresponding image. Because conditioning architectures and mechanisms are always designed based on the target conditional synthesis scenarios, we categorize these works based on their applications, represented by text-to-image and image restoration.

#### 3.1.1 Conditional Models for Text-to-Image(T2I)

Text-to-image is a fundamental task in the field of conditional image synthesis, which establishes the connection between images and the semantic space of text descriptions. Because of the expressiveness of text prompts, text-to-image DMs always serve as the *backbone* for more complicated conditional synthesis tasks including image editing (Kawar et al., 2023; Hertz et al., 2023; Brooks et al., 2023), customization (Gal et al., 2023a; Ruiz et al., 2023), visual signal to image (Mou et al., 2024c; Zhang et al., 2023b), image composition (Yang et al., 2023a) and layout control (Wang et al., 2024b; Li et al., 2023i).

The main challenge in modeling an effective text-to-image framework lies in (a) precisely capturing the users' intention described in text prompts and (b) building the connection between text prompts and images at a acceptable computational cost. In practice, DM-based text-to-image works design different text encoders based on Transformer (Nichol et al., 2022; Rombach et al., 2022), CLIP (Ramesh et al., 2022; Balaji et al., 2022; Gu et al., 2022) or more powerful large language models (Saharia et al., 2022b; Balaji et al., 2022) to extract the features from user provided text prompts. For computational efficiency, these works often train the DMs on a low-dimension space, e.g. compressed latent space (Rombach et al., 2022; Gu et al., 2022) or low-resolution pixel space (Nichol et al., 2022; Saharia et al., 2022c; Balaji et al., 2022; Ramesh et al., 2022), and subsequently convert the synthesized results into desired images via auto-encoders or upsampling diffusion models.

Next, we introduce two representative text-to-image models: Stable Diffusion (Rombach et al., 2022) and Imagen (Saharia et al., 2022b), which serve as the *T2I backbone* for various conditional synthesis tasks.

Similar to VQ-VAE (Van Den Oord et al., 2017) and VQ-GAN (Esser et al., 2021), Stable Diffusion (Rombach et al., 2022) employs a pre-trained autoencoder to compress the generative space into a low-dimensional latent space for computational efficiency. In the training stage, the text-conditioned diffusion model $\boldsymbol{\epsilon}_{\boldsymbol{\theta}}(\mathbf{z}_t,t,\mathbf{c})$ is trained on this latent space to approximate the conditional distribution of the latent representations. In sampling process, the latent representation aligned with given text prompts is firstly generated by the conditional diffusion model on latent space, and then fed into the decoder to recover its corresponding high-quality image.

For conditional control, Stable Diffusion introduces a transformer text encoder to interpret the text prompts and convert into the text embedding. Subsequently text embedding is fused with the features in U-Net architecture of denoising network (Rombach et al., 2022) via cross-attention layers. In practice, the encoder can be different domain-specific experts other than the text encoder. Thereby, Stable Diffusion can be employed into various conditional synthesis scenarios beyond text-to-image.

Following up the pioneer DM-based text-to-image framework GLIDE (Nichol et al., 2022) and Imagen (Saharia et al., 2022b) prefer to train the conditional denoising network on a low-resolution image space and subsequently upsample the synthesized low-resolution image. In order to effectively interpret the complex text prompts, Imagen employs pre-trained large language models (e.g. BERT (Kenton & Toutanova, 2019), GPT (Radford et al., 2021), T5 (Raffel et al., 2020)) as powerful text-encoders. For condition injection,

Imagen (Saharia et al., 2022b) concatenates the encoded text embedding to the key-value pairs of the self-attention layers in denoising network. In Imagen, the basic $64 \times 64$ text-to-image diffusion model is followed by two cascaded super-resolution diffusion models designed to enlarge the resolution of synthesized image from $64 \times 64$ to $1024 \times 1024$.

Recently, the DiT architecture (Peebles & Xie, 2023) has achieved unprecedented sample quality in diffusion-based image synthesis, making it a popular backbone for many cutting-edge diffusion-based text-to-image (T2I) synthesis models (Chen et al., 2023; Esser et al., 2024; Black-Forest, 2024). Building on the conditioning mechanism of Stable Diffusion (Rombach et al., 2022), Cross-DiT (Chen et al., 2023) integrates text conditions into DiT via cross-attention modules. MMDiT (Esser et al., 2024) introduces a novel and scalable DiT-based architecture for T2I synthesis. In contrast to earlier T2I models (Rombach et al., 2022; Ho et al., 2022a; Chen et al., 2023), which rely on consecutive cross-attention and self-attention mechanisms to manage interactions between text prompts and images, MMDiT utilizes a unified self-attention mechanism for bidirectional mixing between text and image tokens. Furthermore, MMDiT (Esser et al., 2024) employs the rectified flow framework to model the transition process from Gaussian distribution to clean images. Leveraging the MMDiT (Esser et al., 2024) framework, Flux (Black-Forest, 2024) achieves state-of-the-art T2I generation performance, excelling in tasks such as handling long sentences and capturing complex multi-object relationships, which remains challenging for Stable Diffusion.

### 3.1.2 Conditional Models for Image Restoration

DM-based conditional training is also widely employed to recover the high-quality clean image $\mathbf{x}$ from a given degraded image $\mathbf{c}$ (Saharia et al., 2022c;a; Ho et al., 2022b; Shang et al., 2024; Zhao et al., 2024a). These studies primarily focus on extracting task-relevant features from degraded images to serve as conditional inputs for supervised training, enabling the model to reconstruct clean images based on these essential representations.

*2.1) Conditioning on degraded images.* The most straightforward modeling approach involves directly conditioning the diffusion model on the degraded image through channel-wise concatenation. A pioneering diffusion-based super-resolution method, SR3 (Saharia et al., 2022c), implements this strategy by concatenating the low-quality reference image with the latent variable in the channel dimension of the U-Net architecture. This straightforward conditioning mechanism allows the U-Net architecture to comprehensively capture the informative content of the low-resolution image. Concurrent work SRDiff (Li et al., 2022a) shifts the generative space of SR3 to the residual space, modeling the difference between paired high- and low-resolution images to avoid reconstructing structures already present in the low-resolution input. As a result, SRDiff achieves performance comparable to SR3 while requiring significantly less computational cost. To adapt SR3 for real-world restoration scenarios, SR3+ (Sahak et al., 2023) introduces second-order degradation simulation to construct more realistic clean/degraded image pairs, thereby enriching the training dataset. Building upon SR3 (Saharia et al., 2022c), CDM (Ho et al., 2022b) proposes a cascaded framework of super-resolution diffusion models to progressively upscale image resolution, while Palette (Saharia et al., 2022a) extends the SR3 framework to a broader range of image restoration tasks through supervised training on task-specific paired datasets.

*2.2) Conditioning on pre-processed features.* However, directly concatenating the degraded image in the channel space imposes a burden on the denoising network, which must implicitly extract task-relevant features from the unprocessed input. To allocate the majority of modeling capacity to task-relevant features, a line of restoration studies (Shang et al., 2024; Zhao et al., 2024a; Jiang et al., 2023a; Xue et al., 2024; Zhang et al., 2024d) advocates first extracting these features from the degraded image and then conditioning the diffusion model on them.

The state-of-the-art super-resolution framework ResDiff (Shang et al., 2024) utilizes a pre-trained Convolutional Neural Network (CNN) to generate a high-quality intermediate image from the initial degraded input. It then conditions the denoising network on this intermediate image, along with its high-frequency components, to model the residual between the intermediate and the clean image. For more challenging restoration tasks, such as underwater image restoration (Zhao et al., 2024a) and low-light image enhancement (Jiang et al., 2023a; Xue et al., 2024), where the input images suffer from severe degradation, a branch of works

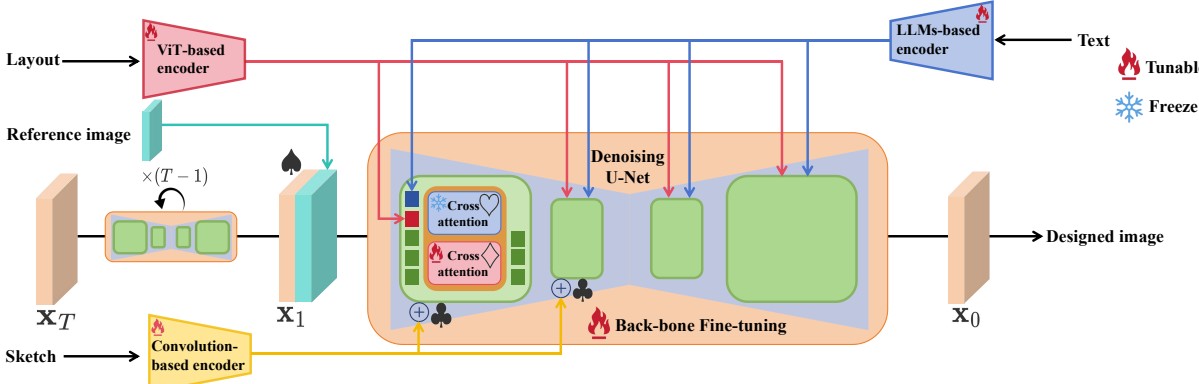

Figure 5: An illustration of the re-purposed denoising network based on the text-to-image backbone, where ♠, ♡, ♣, ◇ denotes condition integration via channel-wise concatenation, T2I attention layers, addition and developed attention modules respectively as described in Sec. 3.2.2.

prefer to condition the model on frequency information extracted by discrete wavelet transformations. To restore real-world text images under severe degradation, DiffTSR (Zhang et al., 2024d) conducts parallel diffusion processes consisting of an image diffusion model for image restoration and a text diffusion model for text recognition. A multi-modal interaction module is further employed to facilitate information exchange between the two diffusion streams.

### 3.1.3 Conditional Models for Other Synthesis Scenarios

While most diffusion model (DM)-based frameworks for complex conditional synthesis scenarios are built by re-purposing text-to-image backbones, a number of studies instead adopt supervised training from scratch for various conditional synthesis tasks. Some of these are early efforts preceding the widespread adoption of DM-based text-to-image models, targeting tasks such as image editing (Preechakul et al., 2022) and visual signal to image synthesis (Wang et al., 2022b; Zhang et al., 2022). Others are developed for novel or highly specialized applications, including medical image synthesis (Li et al., 2023j; Liu et al., 2023a; Moghadam et al., 2023; Meng et al., 2022b), graph-to-image generation (Yang et al., 2022a), and satellite image synthesis (Graikos et al., 2023), where the conditional control signals are difficult to align with the semantic space of text-to-image backbones.

### 3.2 Condition Integration in the Re-purposing Stage

Currently, diffusion models(DMs) are employed in increasingly diverse and complex conditional synthesis scenarios (Ye et al., 2023; Zhang et al., 2023b; Li et al., 2023e; Zhang et al., 2023d; Li et al., 2023i; Wang et al., 2024b; Shi et al., 2024b). Simply training denoising networks from scratch for each conditional synthesis scenario would place a heavy burden on computational resources. Fortunately, pre-trained text-to-image(T2I) DMs effectively associate text embedding with its corresponding image, which serves as a semantic powerful backbone for a wide range of conditional synthesis tasks beyond the T2I. Studies design task-specific denoising network based on T2I backbone and performing fine-tuning on pairs of conditional inputs and image to re-purpose the T2I denoising network to the target task. In practice, the re-purposed denoising network can be divided into three key modules: (a) *Conditional encoder*: The module to encode the task-specific conditional inputs into feature embeddings, (b) *Condition injection*: The module to inject task-related feature embeddings into the T2I backbone, (c) *Backbone*: The T2I backbone that can stay frozen or be fine-tuned during the re-purposing stage. In the re-purposing stage, conditional fine-tuning can be performed in each of these components for condition integration. In this section, we summarize the design choices for these modules adopted by existing works for condition integration during the re-purposing stage.

### 3.2.1 Re-purposed Conditional Encoders

In T2I diffusion models, text embeddings are extracted from input prompts via a text encoder and integrated into the U-Net architecture through cross-attention layers. To adapt the T2I backbone for tasks beyond text-to-image generation, a range of task-specific conditional encoders have been developed to extract features from alternative control signals beyond textual inputs.

*1.1) Convolution layer-based encoder for visual signals.* For visual signals, conditional encoders are mainly designed based on convolution downsample blocks to extract multi-scale structure features.

Pioneer work T2I-Adapter (Mou et al., 2024c) employs a four-layer convolutional network as a lightweight adapter to encode the visual signals into a set of multiscale features. ControlNet (Zhang et al., 2023b) provides a more powerful architecture as the encoder for visual signals, which clones the deep encoding layers in the U-Net architecture of Stable Diffusion. This ControlNet encoder inherits a wealth of prior knowledge in the Stable Diffusion backbone and serves as a deep, robust, and strong architecture for diverse visual signals. Currently, ControlNet delivers state-of-the-art results in diverse visual signal to image tasks and becomes a widely-employed conditional encoder various more complicated conditional synthesis scenarios including explicit lighting control (Kocsis et al., 2024), image composition (Zhang et al., 2023d), image editing (Goel et al., 2023; Zhang et al., 2024e) and virtual try-on (Kim et al., 2024; Zeng et al., 2024).

*1.2) ViT-based encoder for images.* In practice, Vision Transformer (ViT)-based encoders are widely utilized to extract features from conditional control signals represented in the form of images. Since most visual signals can be naturally represented as images, pioneering work such as PITI (Wang et al., 2022a) introduces a ViT-based encoder to project visual inputs into corresponding text embeddings for use in T2I backbones. Similarly, ImageBrush (Yang et al., 2024b) adopts a ViT-based encoder to capture visual editing instructions from paired images before and after editing. Prompt-free Diffusion (Xu et al., 2024) further enhances visual encoding by employing a more powerful context encoder (SeeCoder) built on Swin-L (Liu et al., 2021) to convert given images into meaningful visual embedding.. For customization, a branch of works (Xiao et al., 2023; Ma et al., 2024; Shi et al., 2024a; Gal et al., 2023b; Jia et al., 2023; Li et al., 2023h; Lu et al., 2024; Li et al., 2023a; Shiohara & Yamasaki, 2024) project user-specific objects into features on the textual embedding space via image encoders built upon different ViT-based frameworks such as CLIP (Radford et al., 2021), Swin (Liu et al., 2021), BLIP (Li et al., 2023b), or ArcFace (Deng et al., 2019).

*1.3) LLMs-based encoder for image editing.* In recent years, the rapid development of Multimodal Large Language Models (MLLMs) (Liu et al., 2024b), which are capable of jointly understanding semantic information and associated visual content, has led many recent works to adopt MLLMs as powerful conditional encoders for image editing tasks that require integrating semantic text with the given image for precise manipulation. Fu et al. (2023); Huang et al. (2023c); Li et al. (2023e) leverage trainable Multimodal Large Language Models (MLLMs) (Liu et al., 2024b) module as the encoder for the given source image and editing instruction. Ranni (Feng et al., 2023) used MLLMs to convert description or editing prompts into a semantic panel, which serves as an intermediate representation that contains rich structure and semantic information.

### 3.2.2 Condition Injection

In order to more effectively incorporate information in conditional inputs into the denoising network during the re-purposing stage across various conditional synthesis scenarios, studies in this field have designed different task-specific condition injection approaches to handle different types of conditional control signals. Here, we categorize these methods into the following four categories.

*2.1) Condition injection via concatenation ♠.* For conditional inputs in the form of image, a direct condition injection approach is following the concatenation strategy proposed by SR3 (Saharia et al., 2022c), which concatenates the image form conditional inputs to the latent variable in the channel space of the U-Net architecture. In practice, this conditioning strategy is usually performed with backbone fine-tuning to handle conditional synthesis tasks that involve complex conditional inputs composed of multimodal components, including instruction-based editing (Brooks et al., 2023; Sheynin et al., 2024; Geng et al., 2023) and image composition (Zhang et al., 2023d; Song et al., 2023d; Xie et al., 2023a).

*2.2) Condition injection via T2I attention layers* ♡. In the T2I backbone, the cross-attention layers serve as the conditioning module to inject text embedding into the U-Net architecture. Currently, a branch of works also employs the cross-attention layers in the T2I backbone to inject the features extracted from task-specific conditional encoders (Wang et al., 2022a; Yang et al., 2024b; Xu et al., 2024; Xiao et al., 2023; Gal et al., 2023b; Jia et al., 2023; Li et al., 2023a; Shiohara & Yamasaki, 2024; Zeng et al., 2024).

*2.3) Condition injection via addition* ♣. Because of the alignment between the architecture of conditional encoder and the U-Net encoder in T2I backbone, for convolutional layer-based encoders (Mou et al., 2024c; Zhang et al., 2023b), the extracted features are injected via directly adding these features to the corresponding intermediates layers in the U-Net architecture of T2I backbone.

*2.4) Condition injection via developed attention modules* ◇. To achieve more fine-grained control over the synthesized image, some works developed task-specific attention modules for condition injection in target conditional synthesis scenarios (Ye et al., 2023; Li et al., 2023i; Wei et al., 2023b; Wang et al., 2024b; Mou et al., 2024a).

A branch of works prefers to incorporate extra attention module into the T2I backbone to inject the task-specific conditional control signals (Ye et al., 2023; Wei et al., 2023b; Li et al., 2023i; Hoe et al., 2023; Wang et al., 2024b). IP-adapter (Ye et al., 2023) employs additional image cross-attention layers to inject the image embedding into the T2I backbone. For customization, ELITE (Wei et al., 2023b) leverages two parallel cross-attention layers to inject extracted global and local information of the personal object separately. In T2I backbone, attention layers control the structure and layout information of the synthesized image. To exert accurate object-level layout control, a branch of works prefer to add a trainable attention module between self-attention and cross-attention layers (Li et al., 2023i; Ma et al., 2024; Shi et al., 2024a; Hoe et al., 2023; Wang et al., 2024b). GLIGEN (Li et al., 2023i) adds a gated self-attention layer to the U-Net architecture to inject provided layout information. This conditioning strategy is further employed in customization works (Ma et al., 2024; Shi et al., 2024a) to integrate patch features extracted from personal object images. To perform more detailed layout control, InteractDiffusion (Hoe et al., 2023) designs an attention-based Human-Object Interaction module to inject the interactions between objects. InstanceDiffusion (Wang et al., 2024b) projects different forms of object-level control signals including single points, scribbles, bounding boxes or intricate instance segmentation masks into the feature space through MLP tokenizers, and fuses these features with visual tokens from the text-to-image backbone via gated self-attention layers.

Another line of works modifies the cross-attention mechanism in T2I backbone to achieve more precise control (Qi et al., 2024; Mou et al., 2024a; Lu et al., 2024; Gu et al., 2024). Different from IP-adapter (Ye et al., 2023), DEADiff (Qi et al., 2024) concatenates the key and value attention features derived from image and text embedding respectively and performs a single fused cross-attention mechanism to achieve multimodal conditional control. In practice, performing fused attention mechanism to inject multimodal control signals along with text embedding is also employed in instruct-based editing (Li et al., 2023f) and pose-guided person image synthesis (Lu et al., 2024). To perform local control based on multiple regional prompts, Mix-and-show (Gu et al., 2024) proposes an attention localization strategy in the re-purposing stage, which substitutes the attention map in specified regions with the attention map generated based on the regional prompts.

### 3.2.3 Backbone Fine-tuning

Currently, most of the re-purposing works confine the fine-tuning only on conditional encoders and condition injection modules to ease the computational burden. However, for conditional inputs that contain multimodal components or intricate semantics, performing fine-tuning while freezing the parameters in T2I backbone often fails to fully understand intrinsic connections between the conditional input and target image. In these scenarios, fine-tuning the T2I backbone together with encoders and condition injection modules is a more preferable choice. Based on the fine-tuning strategy, we categorize these works into two types: (a) Fully supervised fine-tuning on annotated datasets, and (b) Self-supervised fine-tuning on bare image dataset.

*3.1) Fully supervised fine-tuning on the annotated dataset.* In practice, we can re-purpose the T2I backbone on the annotated dataset of paired conditional input and image in accordance with the specific task via fully supervised fine-tuning. However, for some synthesis tasks involving complex conditional inputs, a major

difficulty lies in collecting sufficient training data to fine-tune the model (Brooks et al., 2023; Zhang et al., 2023d). For instruct-based editing task, which refers to using instructions instead of text descriptions to guide the editing process, Instructpix2pix (Brooks et al., 2023) provides an effective approach for automatically synthesizing training datasets. Firstly, InstructPix2Pix employs a fine-tuned GPT-3 (Brown et al., 2020) to synthesize editing triplets composed of the input caption, edit instruction and output caption. Subsequently, Instructpix2pix leverages Prompt-to-Prompt (Hertz et al., 2023) to synthesize paired images corresponding to the input captions and output captions, which serves as the paired images before/after editing. This contribution leads to a line of works on DM-based instruction editing. A branch of follow-up works attempts to enhance the T2I backbone in some specific tasks by augmenting the training dataset for target scenario including object removal and inpainting (Yildirim et al., 2023), global editing (Li et al., 2023f), dialog-based editing (Wei et al., 2023a) and continuous editing (Zhang et al., 2024b). InstructDiffusion (Geng et al., 2023) and Emu-edit (Sheynin et al., 2024) fine-tune the T2I backbone on larger and more comprehensive synthesized datasets for a wide range of vision tasks including image editing, segmentation, keypoint estimation, detection and low-level vision. To achieve more accurate editing, Fu et al. (2023); Huang et al. (2023c); Li et al. (2023e) fine-tune the T2I backbone with a more powerful MLLMs-based conditional encoder to enhance the editing prompts. Based on reinforcement learning, HIVE (Zhang et al., 2024c) fine-tunes the instruct-based editing model with a reward model reflecting the human feedback for editing performance.

*3.2) Self-supervised fine-tuning on bare image dataset.* In non-general conditional synthesis scenarios involving image composition or mask-based editing, the form of conditional inputs may be complicated. For example, a classic image composition task aims to fuse a foreground reference image into the background main image within the mask region. In these tasks, collecting annotated training data pairs is almost impossible. A feasible approach is to create paired data based on the target scenario through cropping on a bare image dataset, and thereby fine-tune the T2I backbone in a self-supervised manner. For image composition task, PbE (Yang et al., 2023a) randomly crops the foreground objects from the source image as the reference image and the corresponding mask, while the remaining background as the background main image. Subsequently, PbE (Yang et al., 2023a) fine-tunes the T2I backbone with the cropped reference image and main image. In practice, such a strategy is widely employed in conditional synthesis scenarios involving inpainting (Wang et al., 2023b; Xie et al., 2023a) and composition (Song et al., 2023d; Kim et al., 2023b; Zhang et al., 2023d; Xie et al., 2023b; Chen et al., 2024c). To generate reasonable masks for text-based inpainting, Imagen Editor (Wang et al., 2023b) employs an off-the-shelf object detector to generate masks on the image in captioned image datasets, which covers a region relevant to the text caption of image. SmartBrush (Xie et al., 2023a) randomly augments the cropped training masks to create accurate instance masks, which facilitates the T2I backbone to follow the shape of the input mask in testing-time.

For image composition, the greatest challenge faced by the self-supervised fine-tuning strategy is how to avoid the trivial copy-and-paste solution caused by the training data cropped from a single image (Yang et al., 2023a; Xie et al., 2023b; Zhang et al., 2024e). Currently, image composition frameworks resort to compress the information in the conditional inputs into an information bottleneck. This, in turn, forces the T2I backbone to interpret the intrinsic connections between the conditional input and the desired image, thereby effectively avoiding the copy-and-paste solution. PbE (Yang et al., 2023a) and Dreaminpainter (Xie et al., 2023b) select a part of the image tokens for condition injection to create an information bottleneck. ObjectStitch (Song et al., 2023d) employs a two-stage fine-tuning strategy to decouple the fine-tuning stages of the conditional encoder and the T2I backbone. Zhang et al. (2024e); Chen et al. (2024c); Zhang et al. (2023d) prefer to remove or mask out the information such as colors, textures or background in the source image to prevent identical mapping.

## 3.3 Condition Integration in the Specialization Stage

Although, in theory, any form of conditional input $\mathbf{c}$ can be incorporated into the denoising network $\epsilon_{\boldsymbol{\theta}}(\mathbf{x}_t, t, \mathbf{c})$, in practice, integrating complex control signals into the conditional space of the network during training and re-purposing presents significant challenges. These challenges primarily stem from the difficulty of collecting annotated training data and modeling the intricate relationships between conditional inputs and the desired outputs, thereby limiting the model's capacity to generalize to zero-shot or few-shot conditional inputs.

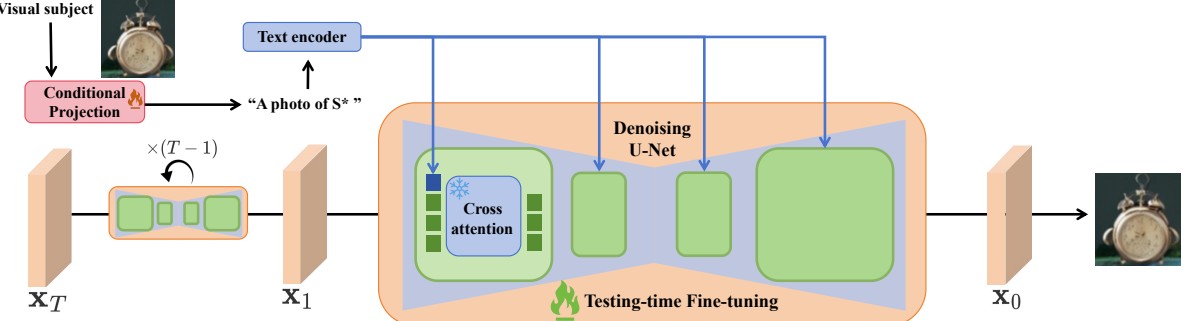

Figure 6: The specialization process to align a given personal object (the clock) with a pesudo-word $S^*$ in the conditional space of a text-to-image backbone. The clock image is cited from Textual Inversion (Gal et al., 2023a).

A straightforward idea to remedy these issues is to align the user-specified conditional inputs with the conditional space of a general T2I backbone through a specialization stage. As shown in Fig. 6, the specialization for given specific conditional inputs is typically achieved by (a) *conditional projection*, which projects the given conditional inputs onto the conditional space of the T2I backbone via embedding optimization (Kawar et al., 2023; Gal et al., 2023a), or Vision-Language Pre-training (VLP) framework (Li et al., 2022b; 2023b), (b) *testing-time model fine-tuning*, which fine-tunes the denoising network to insert the conditional inputs into the prior of the T2I backbone. In practice, works perform condition integration in the specialization stage are mainly targeted to image editing and customization tasks to achieve desired edits on user-specified visual subjects including source image(image editing) and personal objects(customization) while preserving the characteristics and details in these visual subjects (Kawar et al., 2023; Ruiz et al., 2023; Gal et al., 2023a).

### 3.3.1 Conditional Projection

A widely used approach for editing or customization tasks involves projecting the given visual subject into a corresponding text representation within the conditional space of a text-to-image model.

*1.1) Conditional embedding optimization.* In order to find a proper text embedding for given visual subject, a branch of works directly searches the optimal embedding for the user-specified conditional inputs by optimizing the following objective function:

$$\mathbf{v}^* = \arg\min_{\mathbf{v}} \mathbb{E}_{\mathbf{x}=\mathbf{c}_I, \boldsymbol{\epsilon}, t} \left[ \|\boldsymbol{\epsilon} - \boldsymbol{\epsilon}_\theta \left( \mathbf{x}_t, t, \mathbf{v} \right)\|_2^2 \right], \tag{9}$$

where $\mathbf{v}^*$ denotes the optimized text embedding for the user-specified visual subject $\mathbf{c}_I$, and $\boldsymbol{\epsilon}_{\boldsymbol{\theta}}$ denotes the T2I denoising network. The embedding $\mathbf{v}_*$ serves as a pseudo-word $S^*$ for the visual subject and can be further composed into various natural language prompts to create different editing renditions for the given visual subject (Kawar et al., 2023; Gal et al., 2023a).

For image editing, Imagic (Kawar et al., 2023) optimizes the embedding $\mathbf{v}^*$ for the source image. Subsequently, Imagic performs interpolation between optimized source embedding $\mathbf{v}^*$ and target embedding $\mathbf{v}_{tgt}$ to obtain $\overline{\mathbf{v}} = \eta \cdot \mathbf{v}_{tgt} + (1 - \eta) \cdot \mathbf{v}^*$, which serves as the conditional input for denoising process. Diffusion Disentanglement (Wu et al., 2023) optimizes the time-dependent combination weights of the source and target text embeddings along the sampling process instead of interpolation to retrieve time-adaptable embedding for editing. To reduce the computational cost of the optimization process, (Zhang et al., 2023c; Mou et al., 2024b) first employed a image encoder to generate a coarse embedding of the given visual subject, and subsequently fine-tuning the coarse embedding via optimization.

Pioneer customization work Textual Inversion (Gal et al., 2023a) performs optimization to discover the text embedding $\mathbf{v}^*$ for a personal object described by a few reference images (typically 3 to 5). This optimized

embedding $\mathbf{v}^*$ serves as the pseudo-pronoun $S^*$ for the personal object in the further conditional sampling process. To provide human-readable text description instead of text embedding for given personal object, PH2P (Mahajan et al., 2024) employ quasi-newton L-BFGS (Shanno, 1970) to directly optimize discrete tokens from an existing pre-specified vocabulary for the target image.

*1.2) Employing VLP models.* However, performing time-consuming optimization for each new visual subject hinders the deployment of these methods in practical applications. Therefore, a branch of works prefers to employ Vision-Language Pre-training (VLP) models to directly generate the embedding for given visual subjects (Zhang et al., 2023c; Li et al., 2023a).

BLIP (Li et al., 2022b) is a strong VLP framework to synthesize captions for given images, which is widely employed in image editing tasks to generate an initial text prompt to describe the uncaptioned source image (Zhang et al., 2023c; Li et al., 2023a; Bodur et al., 2024; Parmar et al., 2023). BLIP can also be used to enhance user-provided prompts for eliminating editing failure caused by missing contexts in the coarse input prompts (Kim et al., 2023c). Besides, PRedItOR (Ravi et al., 2023) prefer to leverage DALL-E2 (Ramesh et al., 2022) to fuse the source image with the target prompt by performing SDEdit (Meng et al., 2022a) process on the CLIP embedding space.

### 3.3.2  Testing-time Model Fine-Tuning

In editing and customization tasks, directly using the denoising network built through scenario-oriented training and re-purposing often fails to preserve the characteristics and fine details of the user-specified visual subject, primarily due to the absence of subject-specific prior knowledge (Kumari et al., 2023).

To customize the T2I backbone for user-specified conditional inputs, approaches in this category resort to performing testing-time fine-tuning on the T2I backbone to insert the given visual subjects into the denoising network (Ruiz et al., 2023; Kumari et al., 2023).

To better preserve the outlook of the source image in editing tasks, a branch of works (Kawar et al., 2023; Valevski et al., 2023; Zhang et al., 2023c;f) represented by Imagic (Kawar et al., 2023) fine-tune the T2I backbone to bind the source image with its corresponding text description $\boldsymbol{c}_{src}$ in the conditional space of T2I backbone. In order to simultaneously edit the foreground and background in the source image, LayerDiffusion (Li et al., 2023c) employs Segment Anything Model (SAM) (Kirillov et al., 2023) to create masks for foreground objects. Subsequently, LayerDiffusion (Li et al., 2023c) fine-tunes the T2I backbone with a designed loss composed of the diffusion loss in both foreground and background region to edit the foreground object and background independently. SINE (Zhang et al., 2023f) introduces a patch-based fine-tuning strategy which incorporates the positional embedding into conditional T2I space to synthesize arbitrary-resolution edited image.

For the customization task, DreamBooth (Ruiz et al., 2023) fine-tunes the T2I backbone to entangle a fixed unique identifier with the semantic meaning of the personal object. To alleviate the computational burden in the testing-time fine-tuning, followed up works (Kumari et al., 2023; Gal et al., 2023b; Choi et al., 2023; Liu et al., 2023d;e; Gu et al., 2024; Han et al., 2023) prefer to only fine-tune a specific part of model parameters. CustomDiffusion (Kumari et al., 2023) fine-tunes only the cross-attention layers. E4T (Gal et al., 2023b) optimizes low-rank adaptations (LoRA) (Hu et al., 2021) of weight residuals in cross- and self-attention layers to further reduce computational cost. Cones (Liu et al., 2023d) fine-tunes the attention layer concept neurons highly-related to the given visual subject. Cones2 (Liu et al., 2023e) and Mix-and-show (Gu et al., 2024) resort to fine-tune the text encoder in T2I backbone. SVDiff (Han et al., 2023) fine-tunes the singular values of the decomposed convolution kernels.

## 4  Condition integration in the sampling process

In DM-based image synthesis frameworks, the sampling process iteratively reverse noisy latent variable into desired image with the prediction of the denoising network. As mentioned in Sec. 3, integrating the conditional control signals into the denoising network always requires time-consuming training, fine-tuning or optimization. To ease the burden for conditioning the denoising network, numerous works perform condition

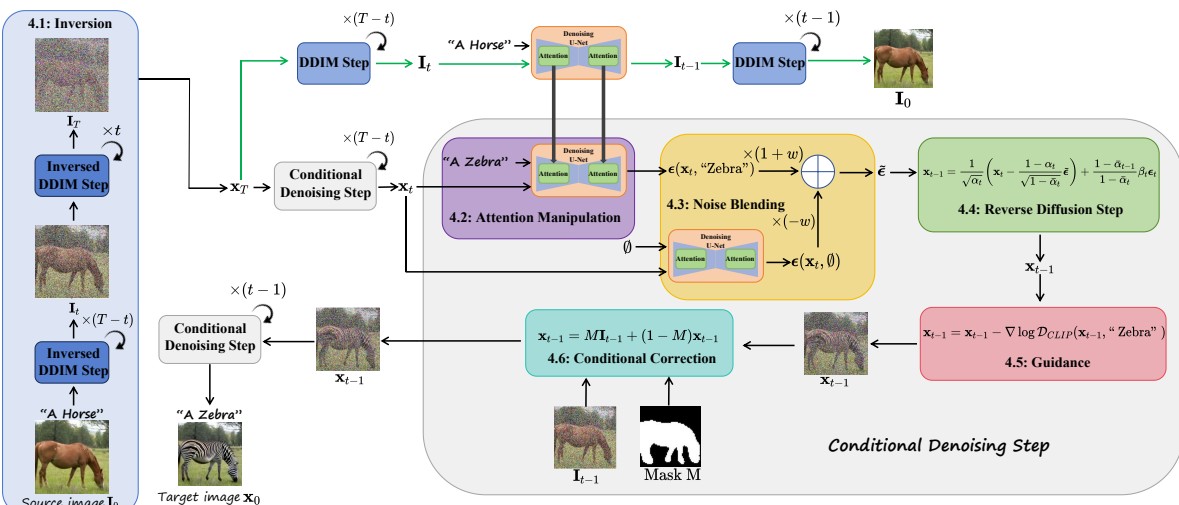

Figure 7: An example of the conditional sampling process for image editing, in which we incorporate all six mainstream in-sampling conditioning mechanisms for diffusion sampling process to provide a comprehensive overview of the content in this section. The sample images are from Diffedit (Couairon et al., 2023).

Table 3: A Comparison of the conditioning mechanisms for sampling process. In the "Inference Cost" column, we present the additional computation cost for performing the corresponding conditioning mechanism in sampling process (where $T$ denotes the total number of sampling steps). The "Guarantee" column illustrates the synthesis quality guarantees of conditioning mechanism. In this column, "theoretical guarantee" indicates the conditioning mechanism is theoretically supported to sample from the corresponding conditional distribution, while "empirical results only" means the method is developed based on successful experimental results, at the cost of disrupting the structure of the standard sampling process.

| Comparison of conditioning mechanisms for sampling process | | | |
|---|---|---|---|
| **Conditioning Mechanism** | **Inference Cost** | **Guarantees** | **Applied scopes** |
| **Inversion** | T NFEs for denoising network | Theoretical guarantee | D,E,F |
| **Attention Manipulation** | T NFEs for denoising network | Empirical results only | E |
| **Noise Blending** | T NFEs for denoising network | Theoretical guarantee | ALL |
| **Revising Diffusion steps** | No additional cost | Theoretical guarantee | B |
| **Guidance** | T NFEs for guidance loss function | Depends on the guidance loss | ALL |
| **Conditional Correction** | No additional cost | Empirical results only | B,D,F |

integration in the sampling process to ensure the consistency between synthesized image and given conditional input without computational intensive supervised-training or fine-tuning (Su et al., 2023; Hertz et al., 2023; Liu et al., 2022; Kawar et al., 2022; Dhariwal & Nichol, 2021; Choi et al., 2021).

Based on how the conditional control signals are incorporated into the sampling process, we divide mainstream in-sampling conditioning mechanisms into six categories: (a) *inversion*, (b) *attention manipulation*, (c) *noise blending*, (d) *revising diffusion process*, (e) *guidance* and (f) *conditional correction*. In Tab. 3, we provides a comparison of the characteristics of all the six conditioning mechanisms for sampling process.

We illustrate these conditioning mechanisms with an exemplary image editing process in Fig. 7. In this section, we will introduce the core idea of these conditioning mechanisms and summarize the corresponding representative works as illustrated in Fig. 8.

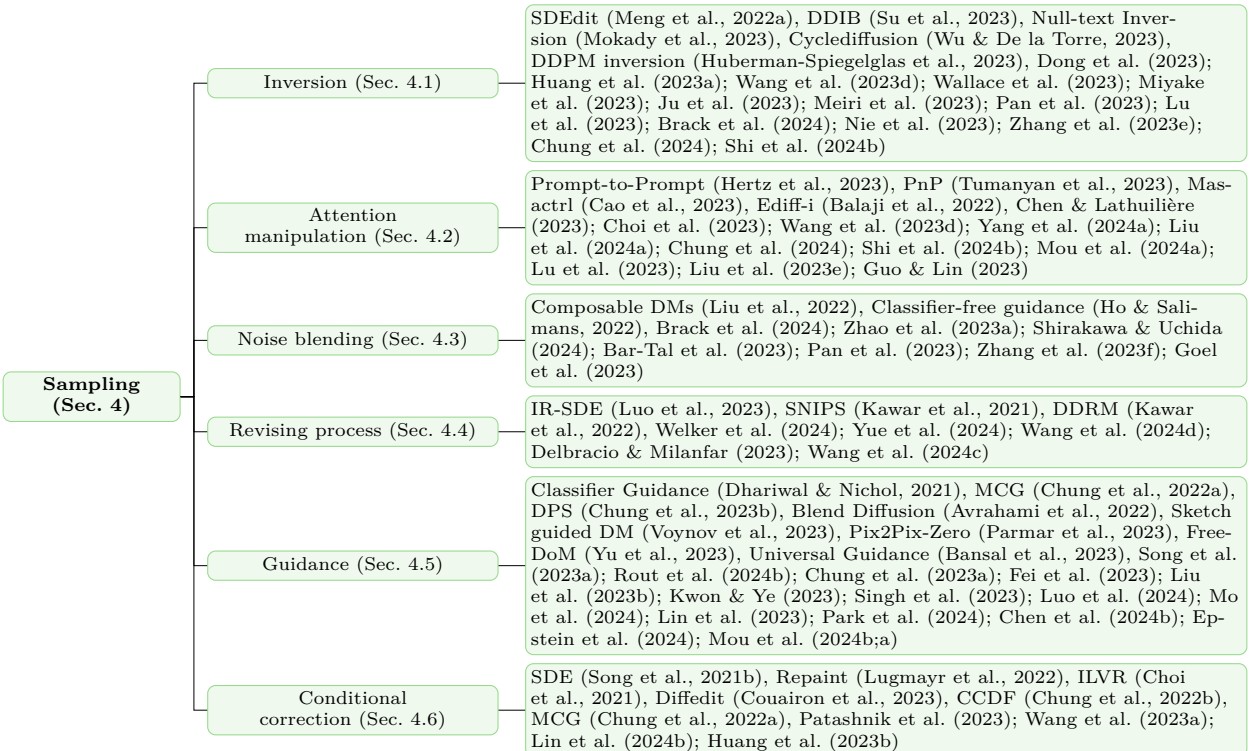

Figure 8: The proposed taxonomy of DCIS works performing condition integration in sampling process.

## 4.1 Inversion

In diffusion model (DM)-based image synthesis, the starting latent variable controls the spatial structure and semantics of synthesized result. Inversion process provides an effective way to encode the given source image back into its corresponding starting latent variable and effectively preserve the image structure and semantics for further editing. In this section, we firstly summarize the inversion approaches in Sec. 4.1.1. Next, we will discuss the applications of inversion in various conditional synthesis scenarios in Sec. 4.1.2.

### 4.1.1 Inversion Approaches

Mainstream inversion approaches perform inversion based on the forward diffusion process, deterministic sampling process and stochastic sampling process. We denote these three basic inversion pathways as *noise-adding inversion*, *deterministic inversion*, and *stochastic inversion*, respectively.

*1.1) Noise-adding inversion.* Noise-Adding Inversion performs a standard forward diffusion process to invert the source image to a certain noise step $T'$, i.e., $q\left(\mathbf{x}_{T'} \mid \mathbf{x}_0\right) = \mathcal{N}\left(\mathbf{x}_{T'}; \sqrt{\bar{\alpha}_{T'}}\mathbf{x}_0, (1 - \bar{\alpha}_{T'})I\right)$, where the latent variable $\mathbf{x}_{T'}$ is a mixture of source image and Gaussian noise.

*1.2) Deterministic inversion.* In practice, noise-adding inversion may smooth out details in the source image. To more precisely preserve image features, deterministic inversion is proposed to encode the source image $\mathbf{x}_0$ into its corresponding latent variable $\mathbf{x}_T$ based on the discretization form of diffusion ODEs such as DDIM (Song et al., 2021a). Theoretically, with a sufficiently large diffusion step $T$, DDIM inversion can guarantee perfect reconstruction, which ensures the latent variable $\mathbf{x}_T$ obtained from DDIM inversion to be a meaningful diffusion starting point encapsulating all features pertaining to the source image $\mathbf{x}_0$.

*1.3) Stochastic inversion.* Different from deterministic inversion approaches performing inversion based on deterministic sampling process, a branch of works prefers to invert the stochastic sampling process in Eq. 3. Unlike the deterministic sampling process, which is determined by the starting point latent variable $\mathbf{x}_T$, the stochastic sampling process involves the noise vector $\boldsymbol{\epsilon}_t$ added in each reverse transition kernel. Therefore, we have to memorize each noise vector $\boldsymbol{\epsilon}_t$ along the inversion process to ensure the reconstruction

property. Despite the additional memory requirements, stochastic inversion alleviates, to some extent, the reconstruction failures caused by accumulated errors in deterministic inversion.

*1.4) Enhanced inversion approaches.* In conditional synthesis, the classifier-free guidance for condition strengthening significantly magnified the accumulated error in inversion process, which leads to poor reconstruction and edit performance. Therefore, a series of inversion methods are developed to ensure the inversion performance under classifier-free guidance.

For deterministic inversion, some approaches prefer to *fine-tune* relevant parameters in the classifier-free guided sampling process to reduce the reconstruction error, including optimizing the null-text embedding (Mokady et al., 2023), text embedding for the source image (Dong et al., 2023), key and value matrix in the self-attention layers (Huang et al., 2023a), and the prompt embedding for cross-attention layers (Wang et al., 2023d). To get rid of the computational burden for fine-tuning, a branch of works has developed *tuning-free* approaches for perfect reconstruction (Wallace et al., 2023; Han et al., 2024; Miyake et al., 2023; Ju et al., 2023). EDICT (Wallace et al., 2023) achieves precise DDIM inversion by utilizing an equivalent reversible process consisting of two coupled noise vectors. Negative-prompt Inversion (Miyake et al., 2023) demonstrates the prompt of the source image can serve as a training-free substitute for null-text embedding. Proxedit (Han et al., 2024) further enhance the reconstruction performance of Negative-prompt Inversion (Miyake et al., 2023) by incorporating a regularization term in classifier-free guidance to prevent over-amplifying the editing direction in sampling process. Fixed-point Inversion (Meiri et al., 2023) and AIDI (Pan et al., 2023) perform fixed-point iterations in each step of DDIM inversion to reduce the accumulation errors due to the discrete DDIM process. Besides, Fixed-point Inversion (Meiri et al., 2023) provides a brief cycle of fixed-point iterations for the VAE-encoded latent representation of the source image to eliminate the misfit between latent representation and the given text prompts in latent diffusion models. TF-ICON (Lu et al., 2023) and LEDITS++ (Brack et al., 2024) perform inversion based on high-order diffusion differential equation solvers (Lu et al., 2022a;b) which significantly accelerates the inversion process and improves the accuracy of inversion.

In theory, for stochastic inversion, any sampling sequence initialized from the source image can serve as the sequence of latent variables in the stochastic inversion process. However, arbitrary sampling trajectories may deviate from the prior marginal distribution of latent variables, thereby compromising the model's editing capability during the reconstruction process. To construct a reasonable sampling sequence, pioneer work Cyclediffusion (Wu & De la Torre, 2023) samples a $\mathbf{x}_T \sim \mathcal{N}(0, \mathbf{I})$ and subsequently denoises it based on the source image $\mathbf{x}_0$ to recover the sampling sequence. DDPM inversion (Huberman-Spiegelglas et al., 2023) constructs an editing-friendly sampling sequence by independently sampling each intermediate latent variable $\mathbf{x}_t$ based on the source image $\mathbf{x}_0$. This approach enables reconstruction of the source image with the noise precisions, effectively mitigating error accumulation during the inversion process. SDE-Drag (Nie et al., 2023) provides a theoretical foundation for the superior editing performance of stochastic inversion compared to its deterministic counterpart. It shows that, under stochastic inversion, the KL divergence between the distribution of the edited image and the prior data distribution decreases, whereas it remains unchanged in widely adopted deterministic inversion methods.

### 4.1.2 Applications of Inversion in Conditional Synthesis

Inversion process converts the provided source image into its corresponding latent variable. In practice, this latent variable can serve as the starting point for the sampling process to perform basic image-to-image translation, text-based image editing or be further manipulated for more complicated tasks.

Image-to-image translation aims to translate the content of a given source image to a desired target appearance, which serves as a fundamental basis for image editing. The pioneering work SDEdit (Meng et al., 2022a) performs this translation by denoising a noise-perturbed version of the source image with the denoising network trained on the target domain. This process effectively preserves the structural content of the source image while imparting it with the visual characteristics of the target domain. Li et al. (2024b) adopt the image-to-image translation strategy introduced in SDEdit to solve the subproblem associated with the prior term in the Half-Quadratic Splitting (HQS) framework (Geman & Yang, 1995) for diffusion-based image restoration.

Based on deterministic inversion, DDIB (Su et al., 2023) proposes a highly flexible framework for image-to-image translation between two data domains, $\alpha$ and $\beta$, via a simple transformation $\mathbf{x}^* = \mathcal{D}_\beta(\mathcal{E}_\alpha(\mathbf{x}))$. Here, $\mathbf{x}$ and $\mathbf{x}^*$ represent the source and target images in domains $\alpha$ and $\beta$, respectively, while $\mathcal{E}_\alpha$ and $\mathcal{D}_\beta$ denote the deterministic inversion and sampling processes performed using diffusion models specific to each domain. The DDIB framework can be applied using either two independently trained diffusion models or a single model conditioned on different control signals.

In practice, text-based image editing, which aims to modify a source image $\mathbf{c}_I$ described by a source prompt $\mathbf{c}_{src}$ to align with the given target prompt $\mathbf{c}_{tgt}$ can be achieved by performing the DDIB image-to-image translation process: $\mathbf{x}^* = \mathcal{D}_{\mathbf{c}_{tgt}}(\mathcal{E}_{\mathbf{c}_{src}}(\mathbf{c}_I))$. Here, $\mathbf{c}_I$ and $\mathbf{x}^*$ represent the paired source and edited images, while $\mathcal{E}_{\mathbf{c}_{src}}$ and $\mathcal{D}_{\mathbf{c}_{tgt}}$ denote the inversion and sampling processes conditioned on the source and target prompts, respectively. However, this editing process can only coarsely preserve semantic consistency and overall structure, often failing to retain the fine-grained details of the source image.

To more effectively preserve intricate details during editing, the inversion process is often augmented with additional conditioning mechanisms. One widely adopted approach involves applying conditional corrections using spatial masks to protect regions that do not require modification (Couairon et al., 2023; Li et al., 2023c; Patashnik et al., 2023; Yang et al., 2024a; Wang et al., 2023a; Lin et al., 2024b; Huang et al., 2023b). Another line of work focuses on manipulating attention features during the editing process to explicitly incorporate the visual appearance of the source image, as discussed in Sec.4.2 (Hertz et al., 2023; Tumanyan et al., 2023). Additionally, some methods enhance source-image fidelity by fine-tuning the model or applying conditional projection techniques during the specialization stage, as outlined in Sec.3.3 (Kawar et al., 2023; Zhang et al., 2023c).

Moreover, by leveraging task-specific conditional encoders that convert multimodal control inputs into text embeddings, the inversion-based editing framework can be extended to conditional synthesis tasks beyond text-driven editing. For instance, InST (Zhang et al., 2023e) performs style transfer by denoising a noise-perturbed reference image obtained via inversion with a denoising network conditioned on embedding vectors extracted from a style image.

For more complex conditional synthesis scenarios, the latent variables obtained through inversion can be further manipulated to incorporate additional information beyond the original source image. In image composition tasks, several methods propose fusing latent representations derived from multiple source images (Chung et al., 2024; Lu et al., 2023). For example, Style Injection in Diffusion (Chung et al., 2024) combines the latent variables of both style and content images obtained via DDIM inversion to achieve style transfer. Similarly, TF-ICON (Lu et al., 2023) performs image composition by integrating the inverted representations of main and reference images. In drag-based editing, the latent variable can be spatially adjusted according to user-provided drag instructions. DragDiffusion (Shi et al., 2024b) refines the latent variable using a motion supervision loss tailored for drag-style manipulation. In contrast, the stochastic inversion-based method SDE-Drag (Nie et al., 2023) employs a copy-and-paste strategy to manipulate latent variables without relying on latent-space optimization.

## 4.2 Attention Manipulation

After determining the starting point via randomly sampling from a Gaussian distribution or inversion methods, the sampling process is performed by iterative denoising steps. As pointed out in E4T (Gal et al., 2023b), the attention layers in the denoising network have the greatest impact on the predicted noise and control the structure and layout of the synthesized image. Therefore, a branch of works resorts to design task-specific manipulation to the attention layers in the denoising network to achieve more accurate control over the spatial layout and geometry of synthesized image (Hertz et al., 2023; Tumanyan et al., 2023; Lu et al., 2023; Patashnik et al., 2023). Different from the works performing fine-tuning on modified attention module in re-purposing stage (Li et al., 2023i; Ye et al., 2023), approaches in this category manipulate the attention layers via tuning-free replacement or localization during the sampling process.

### 4.2.1 Replacement Manipulation

Pioneer attention manipulation works are designed to preserve the structure of the source image during the inversion-based image editing process. Prompt-to-Prompt (Hertz et al., 2023) performs parallel sampling processes for the inverted source image separately conditioned on source and target prompts. During the sampling process, Prompt-to-Prompt replaces the cross-attention maps in the editing branch with its counterpart in the reconstruction branch to preserve the structure of the source image during the editing process. This replacement strategy is further employed in follow-up works for face aging editing (Chen & Lathuilière, 2023) and customization-based editing (Choi et al., 2023). P2Plus (Wang et al., 2023d) further performs attention replacement when predicting unconditional noise term in Eq. 7 to achieve more accurate editing under classifier-free guidance. To prevent undesired changes caused by cross-attention leakage, DPL (Yang et al., 2024a) optimizes the word embedding corresponding to the noun words in the source prompt to produce more suitable cross-attention maps for attention replacement.

PnP (Tumanyan et al., 2023) points out that more detailed spatial features are restored in self-attention layers compared to the cross-attention maps. Therefore, a branch of editing works (Tumanyan et al., 2023; Liu et al., 2024a; Cao et al., 2023) prefers to replace query and key features in self-attention layers to achieve better structure preservation. This replacement strategy is followed by works designed for drag-based editing (Shi et al., 2024b; Mou et al., 2024a) and style transfer (Chung et al., 2024) to ensure the consistency between the synthesized result and the provided source image. However, performing replacement manipulation on attention maps locks the spatial layout of the generated image to that of the source image. To support more complex image editing scenarios, a series of works (Cao et al., 2023; Huang et al., 2024a) prefer to perform attention replacement on the key and value features within the attention layers, enabling the model to handle structural changes in non-rigid editing tasks.

In practice, the effectiveness of editing highly depends on the capability of the underlying text-to-image model. Currently, DiT-based text-to-image models (Peebles & Xie, 2023; Chen et al., 2023; Esser et al., 2024; Black-Forest, 2024) demonstrate significantly stronger language understanding and image generation capabilities compared to traditional text-to-image models (Rombach et al., 2022; Ho et al., 2022a). As a result, a series of recent works (Wang et al., 2024a; Tewel et al., 2024) have chosen to apply the classic attention manipulation strategies to these models (Esser et al., 2024; Black-Forest, 2024), which achieves state-of-the-art performance in image editing.

### 4.2.2 Localization Manipulation

In order to enable more precise layout control in the synthesized image, a branch of works manipulates (Patashnik et al., 2023; Lu et al., 2023; Balaji et al., 2022). the attention layers with masks or segmentation maps that specify object locations.

Some of these works propose localized self-attention mechanisms to address different regions separately and locate the contents into desired regions. Masactrl (Cao et al., 2023) and Object-Shape Variation (Patashnik et al., 2023) firstly extract the regions with attention value above a threshold in the cross-attention maps for object text tokens as foreground masks. Subsequently, Masactrl performs self-attention for foreground and background separately to prevent mixing the foreground objects and the background. Object-Shape Variation (Patashnik et al., 2023) restricts the region for attention replacement on the background instead of injecting the full self-attention maps in every denoising step. For image composition, TF-ICON (Lu et al., 2023) fuses the attention features extracted from the reconstruction process of both main and reference images via a cross-attention mechanism to create a composite self-attention map seamlessly blending the two images.

Another line of work incorporates an increment into the cross-attention map to adjust the attention values in the region for designated objects and thereby achieve layout control for the synthesized image. Pioneer text-to-image work Ediff-i (Balaji et al., 2022) successfully guides the object described by the nouns in the text prompt to the specified area by enhancing the its attention values in the corresponding region. Similarly, Cones2 (Liu et al., 2023e) increases the attention values in the region corresponding to desired objects while reducing the attention values in irrelevant regions to perform layout control. For image editing, FoI (Guo &

Lin, 2023) amplifies the attention value in the region of the foreground object to be edited to achieve more precisely control for the objects in accordance with editing instructions.

### 4.3 Noise Blending

The noise blending process integrates noise predictions from multiple (conditional) diffusion models to enable a unified sampling process guided by multiple control signals. Based on the corresponding application scenarios, existing noise blending methods can be categorized into Noise Composition and Classifier-Free Guidance.

#### 4.3.1 Noise Composition

In conditional synthesis scenarios that require generating images based on multiple control signals, directly training a denoising network to simultaneously handle all conditional inputs often results in prohibitively high training costs. A widely adopted approach for handling multi-conditional synthesis involves independently predicting the noise $\boldsymbol{\epsilon}_i$ for each conditional component $\mathbf{c}_i$, and subsequently composing them into a proxy noise $\tilde{\boldsymbol{\epsilon}}$ that reflects the influence of all control signals. Composable Diffusion Models (Liu et al., 2022) propose a noise composition method based on Bayes' rule to enable multi-conditional image synthesis, formulated as follows::

$$\tilde{\boldsymbol{\epsilon}} = \boldsymbol{\epsilon}_\theta\left(\mathbf{x}_t, t\right) + \sum_{i=1}^{n} w_i \left(\boldsymbol{\epsilon}_\theta\left(\mathbf{x}_t, t, \mathbf{c}_i\right) - \boldsymbol{\epsilon}_\theta\left(\mathbf{x}_t, t\right)\right), \tag{10}$$

where the unconditional denoising network $\boldsymbol{\epsilon}_\theta\left(\mathbf{x}_t, t\right)$ can be trained along with the conditional model by substituting the conditional inputs $\mathbf{c}$ with emptyset $\emptyset$. The noise composition can be performed based on masks or layouts to locate the objects in provided conditional inputs into desired regions. To perform image editing on multiple instructions, LEDITS++ (Brack et al., 2024) calculates the mask for the region related to each instruction with the grounding information in cross-attention layers and noise estimations. Subsequently, LEDITS++ (Brack et al., 2024) performs noise composition based on the formula of Eq.10 while restricting effect of the conditional term $\boldsymbol{\epsilon}_\theta\left(\mathbf{x}_t, t, \mathbf{c}_i\right) - \boldsymbol{\epsilon}_\theta\left(\mathbf{x}_t, t\right)$ of each editing instruction $\mathbf{c}_i$ in its corresponding mask region. In order to fuse the synthesized results of two diffusion models, MagicFusion (Zhao et al., 2023a) firstly generates a mask by contrasting the saliency map of the two diffusion models to differentiate the region controlled by each model. Subsequently, MagicFusion (Zhao et al., 2023a) settles the noise into the region controlled by its corresponding diffusion model. Similarly, NoiseCollage (Shirakawa & Uchida, 2024) independently estimates the noises for each individual object and then merges them with a crop-and-merge operation based on the provided layouts. In order to perform more seamless noise composition, Multi-diffusion (Bar-Tal et al., 2023) blends the noise by solving an optimization objective with closed-form optimal solution, which ensure the consistency in composed noise map $\tilde{\boldsymbol{\epsilon}}$.

#### 4.3.2 Classifier-Free Guidance

As described in Sec.2.4, classifier-free guidance (Ho & Salimans, 2022) performs extrapolation noise blending between the conditional noise prediction and the unconditional noise prediction $\tilde{\boldsymbol{\epsilon}}_\theta\left(\mathbf{x}_t, \mathbf{c}\right) = (1+w)\boldsymbol{\epsilon}_\theta\left(\mathbf{x}_t, \mathbf{c}\right) - w\boldsymbol{\epsilon}_\theta\left(\mathbf{x}_t\right)$ to balance the quality and diversity of synthesized samples. In practice, some works also propose variations of classifier-free guidance for condition incorporation in different conditional synthesis scenarios. Instructpix2pix (Brooks et al., 2023) and Pairdiffusion (Goel et al., 2023) develop the classifier-free guidance to adjust the conditioning strength for each component in multiple conditional inputs by decomposing the multi-conditional score function. For customization tasks, SINE (Zhang et al., 2023f) interpolates the noise prediction on specialized and pre-trained model to obtain conditional noise prediction in classifier-free guidance, which alleviates the overfitting in the specialized model. Null-text Guidance perturbs the classifier-free guidance by altering the noise-level in unconditional prediction to smooth out some realistic details and create cartoon-style images. For inversion-based editing, AIDI (Pan et al., 2023) proposes a blended classifier-free guidance based on the positive/negative masks indicating the area to be edited or preserved, which enables larger guidance scales and ensures more accurate editing results.

### 4.4 Revising Diffusion Process

Most of in-sampling conditioning mechanisms such as Guidance, Conditional Correction and Attention Manipulation performs modification on the standard formulation of the denoising step, which leads to deviations from the predetermined sampling trajectory and results in artifacts in synthesized images. Therefore, a branch of works prefer to incorporate the condition control signals into the denoising step via revising the formulation of standard diffusion process to adapt the conditional synthesis task (Luo et al., 2023; Yue et al., 2024; Kawar et al., 2022; Wang et al., 2024c). Thereby, the conditional control signals can be incorporated into the corresponding reverse diffusion step of the revised diffusion process without deviations from the diffusion formulation.

Based on the revision on diffusion process, these works can be divided into two categories: (a) *mean-reverting SDEs*, which revise the diffusion process to preserve the information in conditional inputs for image restoration, (b) *decomposition-based noise redefinition*, which incorporate a sequence of additive noises in the sampling process on the spectral space to revise the noise-level mismatch in noisy linear problems.

#### 4.4.1 Mean-Reverting SDEs

In numerous restoration tasks, most structure and semantic features of the target image are provided by the degraded image $\mathbf{c}$. To avoid consuming part of the model capability on regenerating these features from pure Gaussian noise, numerous studies design novel diffusion processes in which the diffused output $\mathbf{x}_T$ approximates a noisy version of degraded image $\mathbf{c}$ instead of pure Gaussian noise (Welker et al., 2024; Luo et al., 2023; Yue et al., 2024; Wang et al., 2024d; Delbracio & Milanfar, 2023). IR-SDE (Luo et al., 2023) constructs a set of mean-reverting SDEs identified by degraded image $\mathbf{c}$, which models the diffusion process from clean image $\mathbf{x}$ to a Gaussian distribution averaged on degraded image. Subsequently, IR-SDE trains a conditional denoising network to predict the score function in the reversed mean-reverting SDEs to recover the clean image from the noisy degraded image. Similarly, ResShift (Yue et al., 2024) and DriftRec (Welker et al., 2024) construct an iterative degradation process from a high-resolution image to its corresponding low-resolution image as the diffusion process and train a conditional denoising network to reverse the degradation process for super-resolution. SinSR (Wang et al., 2024d) distills the sampling process of ResShift (Yue et al., 2024), thereby achieving one-step DM-based super-resolution. InDI (Delbracio & Milanfar, 2023) constructs a continuous forward degradation process derived from interpolation: $\mathbf{x}_t = (1-t)\mathbf{x}+t\mathbf{c}$ and trains a denoising network on paired clean/degraded images to predict the clean image $\mathbf{x}_0$. Subsequently, image restoration can be performed by reversing the interpolation-based degradation process with the prediction of this denoising network.

#### 4.4.2 Decomposition-Based Noise Redefinition

Methods in this category construct novel diffusion processes to recover image $\mathbf{x}$ from its partial measurement $\mathbf{c}$ in the noisy linear inverse problems as follows $\mathbf{c} = \boldsymbol{H}\mathbf{x}+\mathbf{n}$, where $\boldsymbol{H}$ is a known linear degradation matrix, $\mathbf{n} \sim \mathcal{N}\left(0, \sigma_{\mathbf{c}}^2\mathbf{I}\right)$ is an i.i.d. additive Gaussian noise with known variance. In practice, numerous restoration tasks including inpainting, super-resolution and colorization can be written in the form of this noisy linear inverse problems. SVD Decomposition-based methods firstly perform SVD decomposition on the linear degradation matrix $\mathbf{H}$ to decouple the components in the measurement $\mathbf{c}$. Thereby, the components in measurement $\mathbf{c}$ on spectral space can be viewed as a noisy version of their counterparts derived from clean image $\mathbf{x}$. In order to incorporate the measurement $\mathbf{c}$ into the diffusion process while preventing the mismatch in noise-level caused by the noise in measurement $\mathbf{c}$, decomposition-based methods design a proper noise sequence to link the noise in the measurement $\mathbf{c}$ with the noise added in the standard diffusion process. It can be proven that the optimized unconditional denoising network pre-trained on the prior of the clean image $\mathbf{x}$ is also the optimal solution for the variational objective of the designed novel diffusion process. Thereby, we can perform the sampling process in the spectral space to recover clean image $\mathbf{x}$ from its noisy counterpart $\mathbf{c}$ based on a pre-trained unconditional denoising network. SNIPS (Kawar et al., 2021) and DDRM (Kawar et al., 2022) construct SVD decomposition-based novel diffusion process in the spectral space based on the annealed Langevin dynamics framework provided by NCSN (Song & Ermon, 2019) and the Markov chain diffusion process provided by DDPM (Ho et al., 2020) respectively.

Different from SNIPS and DDRM, DDNM (Wang et al., 2024c) construct a general solution $\hat{\mathbf{x}}$ based on range-null space decomposition which holds $\mathbf{H}\hat{\mathbf{x}} \equiv \mathbf{c}$. In each denoising step, DDNM (Wang et al., 2024c) projects the denoising output $\mathbf{x}_{0|t}$ onto the general solution to guarantee the consistency between denoising output $\mathbf{x}_{0|t}$ and given measurement $\mathbf{c}$. For noisy linear inverse problem $\mathbf{y} = \boldsymbol{H}\mathbf{x} + \mathbf{n}$, DDNM (Wang et al., 2024c) incorporates a scaling factor into the formulation of the general solution and designs a noise sequence corresponding to the scaling factor during sampling process to ensure the noise level in $\mathbf{x}_{t-1}$ aligned with the definition of $q(\mathbf{x}_{t-1} \mid \mathbf{x}_0)$ for the pre-trained unconditional denoising network.

## 4.5 Guidance

Sampling from the conditional distribution $p(\mathbf{x}|\mathbf{c})$ with diffusion models requires approximating the conditional score function $\nabla_{\mathbf{x}_t} \log p_t(\mathbf{x}_t \mid \mathbf{c})$ with a conditional denoising network $\boldsymbol{\epsilon_\theta}(\mathbf{x}_t, t, \mathbf{c})$. sssIn practice, guidance provides another pathway to approximate the conditional score function without time-consuming conditional training, since the conditional score function can be decomposed into an unconditional score function and the gradient of log likelihood as follows:

$$\nabla_{\mathbf{x}_t} \log p_t(\mathbf{x}_t \mid \mathbf{c}) = \nabla_{\mathbf{x}_t} \log p_t(\mathbf{c} \mid \mathbf{x}_t) + \nabla_{\mathbf{x}_t} \log p_t(\mathbf{x}_t), \tag{11}$$

where the score function $\nabla_{\mathbf{x}_t} \log p_t(\mathbf{x}_t)$ can be estimated by an unconditional denoising network $\boldsymbol{\epsilon_\theta}(\mathbf{x}_t, t)$. Guidance-based methods design task-specific guidance losses to reflect the alignment between intermediate latent variable $\mathbf{x}_t$ and conditional inputs $\mathbf{c}$ at each time step, which approximates the log likelihood $\log p_t(\mathbf{c} \mid \mathbf{x}_t)$. For multiple conditional inputs, guidance can also be employed to perform conditional control for part of the conditional inputs. We can split the conditional inputs $\mathbf{c}$ into components $\mathbf{c}_0$ and $\mathbf{c}_1$ which are incorporated into the diffusion synthesis framework with conditional denoising network and guidance. In this case, the conditional score function can be written as $\nabla_{\mathbf{x}_t} \log p_t(\mathbf{x}_t \mid \mathbf{c}_0, \mathbf{c}_1) = \nabla_{\mathbf{x}_t} \log p_t(\mathbf{c}_1 \mid \mathbf{x}_t, \mathbf{c}_0) + \nabla_{\mathbf{x}_t} \log p_t(\mathbf{x}_t \mid \mathbf{c}_0)$, where $\nabla_{\mathbf{x}_t} \log p_t(\mathbf{x}_t \mid \mathbf{c}_0)$ can be estimated by a denoising network conditioned on $\mathbf{c}_0$ and $\log p_t(\mathbf{c}_1 \mid \mathbf{x}_t, \mathbf{c}_0)$ can be estimated by the guidance loss.

Currently, by designing different task-specific guidance losses, guidance-based methods are widely adopted across various conditional synthesis scenarios. Apart from the simplest case discussed in Sec.2.4, where a classifier is trained conditioned on class labels(Dhariwal & Nichol, 2021), training an accurate classifier becomes challenging when dealing with more complex control signals. To address this, subsequent works have proposed more flexible guidance losses that do not require explicit training or optimization. In the following, we categorize these approaches based on their target applications.

### 4.5.1 Guidance for Inverse Problems

As mentioned in Sec. 4.4.2, a wide range of restoration tasks can be expressed by recovering a clean image $\mathbf{x}$ from a given partial measurement $\mathbf{c}$ in the form of noisy inverse problems: $\mathbf{c} = \mathcal{A}(\mathbf{x}) + \boldsymbol{n}, \; \boldsymbol{n} \sim \mathcal{N}(0; \sigma_\mathbf{c}^2 \boldsymbol{I})$, where $\mathcal{A}$ is a known degradation function and $\boldsymbol{n}$ denotes the additive noise. In practice, approximating the likelihood $p_t(\mathbf{c}|\mathbf{x}_t)$ and performing guidance on the sampling process is a widely employed strategy to solve the noisy inverse problem. Fig. 9 provides an illustration of the sampling process with guidance for the inverse problem.

MCG (Chung et al., 2022a) and DPS (Chung et al., 2023b) approximate the gradient of likelihood as: $\nabla_{\mathbf{x}_t} \log p_t(\mathbf{c} \mid \mathbf{x}_t) \approx \nabla_{\mathbf{x}_t} \log p(\mathbf{c}|\mathbf{x}_{0|t}) = -\frac{1}{\sigma_\mathbf{c}^2} \nabla_{\mathbf{x}_t} \left\| \mathbf{c} - \mathcal{A}(\mathbf{x}_{0|t}) \right\|_2^2$. The estimation error can be proven to converge to 0 as $\sigma_\mathbf{c} \to \infty$ in

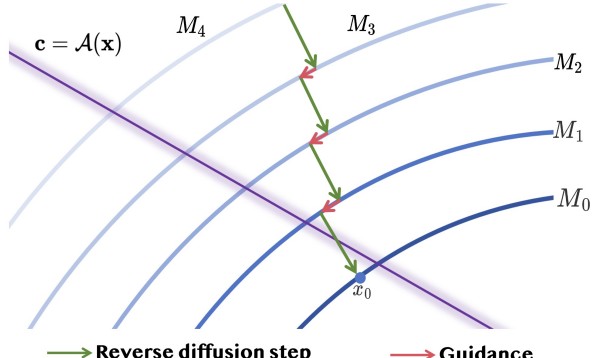

Figure 9: An illustration of the guided sampling process for inverse problems. The curve $M_t$ denotes the data manifold of the intermediate diffused output $\mathbf{x}_t$. The guidance process (red arrow) moves $\mathbf{x}_t$ towards the data manifold satisfying the constraint $\mathbf{c} = \mathcal{A}(\mathbf{x})$ (purple line).

most inverse problems. IIGDM (Song et al., 2023a) provides a more accurate estimation for the likelihood by approximating $p_t (\mathbf{x}_0 \mid \mathbf{x}_t)$ with a Gaussian distribution averaged on $\mathbf{x}_{0|t}$.

To perform these guidance approaches for inverse problems on latent diffusion models (Rombach et al., 2022), PSLD (Rout et al., 2024b) adds an additional guidance term measuring the reconstruction ability of the intermediate denoising output $\mathbf{z}_{0|t}$ to avoid guiding the sampling trajectory towards latent variable $\mathbf{z}_0$ away from the manifold of real data. Resample (Song et al., 2024) introduces a stochastic resampling schema that reliably maps the measurement-guided intermediate diffuse output $\mathbf{x}_{0|t}$ to the latent variable $\mathbf{x}_{t-1}$ for the subsequent sampling step, effectively preventing noisy image reconstructions in latent diffusion models.

However, these guidance approaches can only estimate the likelihood term in inverse problems with known a concrete form of the degradation operator $\mathcal{A}(\cdot)$. This hinders the deployment of these approaches for unknown real-world degradations. BlindDPS (Chung et al., 2023a) explores the applicability of DPS to blind inverse problems, in which the degradation operator $\mathcal{A}_{\boldsymbol{\varphi}}(\cdot)$ is parameterized with unknown parameter $\boldsymbol{\varphi}$. To identify the degradation parameter along with the sampling process for the desired image, BlindDPS trains a diffusion model for the parameters $\boldsymbol{\varphi}$ in the degradation operator. In the sampling process, BlindDPS employed a similar approximation strategy as DPS (Chung et al., 2023b) to estimate the likelihood term as follows:

$$p_t (\mathbf{c} \mid \mathbf{x}_t, \boldsymbol{\varphi}_t) \approx p (\mathbf{c} \mid \mathbf{x}_{0|t}, \boldsymbol{\varphi}_{0|t}). \tag{12}$$

Subsequently, BlindDPS performs a parallel sampling process to simultaneously recover the clean image $\mathbf{x}$ and the unknown degradation parameters $\boldsymbol{\varphi}$ from the conditional distribution $p(\mathbf{x}, \boldsymbol{\varphi}|\mathbf{c})$ with the estimated likelihood in Eq. 12. GDP (Fei et al., 2023) offers a heuristic approximation for the likelihood term, which consists of a distance metric measuring the consistency to conditional inputs and an optional quality enhancement loss to control some desired properties in synthesized results. GDP can also be employed in blind inverse problems by optimizing the degradation parameters in the degradation function $\mathcal{A}$ with the distance metric during sampling process.

### 4.5.2 Guidance for Semantic Control

Guidance can also be employed to ensure the consistency to provided semantic control signals, such as text prompts or semantic images, without time-consuming fine-tuning or training. Semantic guidance losses are usually designed based on pre-trained CLIP models which have a rich shared image-text embedding space.

Blend Diffusion (Avrahami et al., 2022) aims to inpaint the masked region $\mathbf{c}_m$ in source image $\mathbf{c_I}$ based on the provided text description $\mathbf{c}_d$. It designs a CLIP loss for the conditional inputs $\mathbf{c} = (\mathbf{c}_m, \mathbf{c}_I, \mathbf{c}_d)$ as follows:

$$L(\mathbf{x}_t, \mathbf{c}) = \mathcal{D}_{CLIP} (\mathbf{x}_{0|t}, \mathbf{c}) + \lambda \mathcal{D}_{bg} (\mathbf{x}_{0|t}, \mathbf{c}), \tag{13}$$

where $\mathcal{D}_{CLIP}$ measures the CLIP distance between the intermediate denoising output $\mathbf{x}_{0|t}$ and text description $\mathbf{c}_d$ in mask region for semantic alignment, and $\mathcal{D}_{bg}$ calculates the MSE and LPIPS similarity between $\mathbf{x}_{0|t}$ and source image $\mathbf{c_I}$ in unmasked region for the faithfulness to source image. To jointly control the sampling process using both a text prompt and a style reference image, SDG (Liu et al., 2023b) employs a linear combination of the CLIP distances between the current denoising output and the embeddings of both the text and the reference image as the guidance loss. In addition, DiffuseIT (Kwon & Ye, 2023) introduces an auxiliary structure loss computed from the self-attention features of the source image extracted using a Vision Transformer (ViT) (Dosovitskiy et al., 2020), in order to better preserve the structural integrity of the source image.

### 4.5.3 Guidance for Visual Signals

In practice, a branch of works employs guidance to ensure the consistency between the diffused output and the given visual signal. In order to measure the consistency between intermediate diffused output and the provided visual signal, some works train neural networks to project the intermediate diffused output $\mathbf{x}_t$ onto its corresponding visual signal and leverage a distance metric as the guidance loss for sketch-to-image (Voynov et al., 2023) and stroke-to-image (Singh et al., 2023). Readout Guidance (Luo et al., 2024) provide a unified guidance-based framework for diverse visual signals to image task by training various

readout heads to synthesize different task-specific visual feature maps reflecting the spatial layout or inherent correspondence in images to perform guidance. Different from these works, FreeControl (Mo et al., 2024) prefers to impose guidance loss on the difference in the space of PCA components of self-attention map between the intermediate diffused output and visual signal.

### 4.5.4 Guidance for Attention Layers

In DM-based conditional image synthesis, the attention layers in denoising network effectively control the layout, structure and semantics of synthesized image. However, directly manipulating the attention layers through replacement or localization as described in Section 4.2 introduces artificial modifications to the internal parameters of the denoising network and may impair its modeling capability. Therefore, a branch of works employ guidance to achieve soft control for attention layers.

For image editing, attention guidance is performed as a substitution of attention replacement to softly control the consistency between source image and edited result. Pix2Pix-Zero (Parmar et al., 2023) employs a guidance loss measuring the $L_2$ distance between the cross-attention maps in editing branch and reconstruction branch instead of the replacement manipulation in Prompt-to-prompt (Hertz et al., 2023). In order to find a more expressive attention map as a guidance reference, Rediffuser (Lin et al., 2023) employs a sliding fusion strategy to fuse the cross-attention maps obtained from sampling branches conditioned on source prompt, target prompt and an intermediate representation. EBMs (Park et al., 2024) employs an energy function to guide the integration of the semantic information in editorial prompts with the structure and layout of source image restored in cross-attention layers.

Attention guidance can also be employed to perform attention localization. For object-level layout control, Chen et al. (2024b) employs guidance to control the cross-attention map, which locates the objects in text prompts into their desired bounding boxes. Self-guidance (Epstein et al., 2024) extracts various characteristics including position, size, shape and appearance of the desired object from the intermediate activations and attention maps. Subsequently, Self-guidance places constraints on these characteristics with guidance loss measuring their consistency to desired conditional control signal. For drag-based editing tasks which target to move certain foreground contents in source image into target region, Dragondiffusion (Mou et al., 2024a) designs energy functions based on the cosine distance between intermediate features in the U-Net decoder as guidance to ensure correspondence between the original content region and the target dragging region. DiffEditor (Mou et al., 2024b) develops the guidance framework of DragonDiffusion (Mou et al., 2024a) by introducing SDE-based sampling process on the masked region instead of ODEs to improve editing flexibility.

### 4.5.5 Enhanced Guidance Framework

In some complicated conditional synthesis scenarios, simply incorporating the gradient of guidance loss in each denoising step may lead to artifacts and strange behaviors because of the failure in balancing the realness and guidance constraint satisfaction in guided sampling process. Therefore, some state-of-the-art guidance works provide enhanced guidance frameworks to more effectively fuse the prior knowledge in pre-trained models and the information in conditional control signals. FreeDoM (Yu et al., 2023) employs a time-travel strategy that rolls back the intermediate latent variable $\mathbf{x}_t$ to a certain previous time step $\mathbf{x}_{t+j}$ and resamples it to time step $t$ again. This strategy inserts additional steps into the guided sampling process, allowing for a more seamless integration of the information from the pre-trained model and the conditional control signals.

In order to enhance the sample consistency to conditional control signals, a branch of works (Bansal et al., 2023; Zhu et al., 2023; Song et al., 2024; Zhang et al., 2024a) performs a multi-step gradient descent optimization process instead of the traditional one-step gradient guidance to find the point with minimum guidance loss in the vicinity of the intermediate denoising output $\mathbf{x}_{0|t}$, and adopts this point to infer the next latent variable $\mathbf{x}_{t-1}$. Universal Guidance (Bansal et al., 2023) refers to this enhanced guidance framework as backward guidance, and it has successfully generated quality images in tasks such as segmentation, face recognition, object detection, and classifier signals. For inverse problems, DAPS (Zhang et al., 2024a) guides the intermediate denoising output $\mathbf{x}_{0|t}$ toward the conditional distribution $p(\mathbf{x}_0 \mid \mathbf{x}_t, \mathbf{c})$ through multi-step MCMC sampling methods (Welling & Teh, 2011). DiffPIR (Zhu et al., 2023) utilizes a Half-Quadratic

Splitting (Geman & Yang, 1995) (HQS) optimization process as an optimization-based guidance to ensure the consistency to the partial measurement **c**.

TFG(Ye et al., 2024) integrates several classic diffusion guidance approaches (Chung et al., 2023b; Yu et al., 2023; Bansal et al., 2023; He et al., 2024; Song et al., 2023b) into a unified framework and optimizes the associated hyper-parameters via an efficient beam search strategy, leading to enhanced performance in diverse conditional synthesis tasks.

## 4.6 Conditional Correction

In some conditional synthesis scenarios, the synthesized images are controlled by the constraints specified by conditional inputs **c** (such as the formulation of inverse problems). To ensure the synthesized result to be consistent to the inputs **c**, conditional correction-based methods perform a correction operator on the intermediate diffused output $\mathbf{x}_t$ (or $\mathbf{x}_{0|t}$), which projects the current diffused output onto the data manifold satisfying the constraint imposed by given conditional control signal **c**. Subsequently, this corrected latent variable will be passed into next denoising step, as shown in Fig. 10.

Currently, conditional correction are widely employed in image inpainting tasks, which involves synthesizing content for the masked region $\mathbf{c}_m$ in incomplete reference image $\mathbf{c}_y$. The constraint

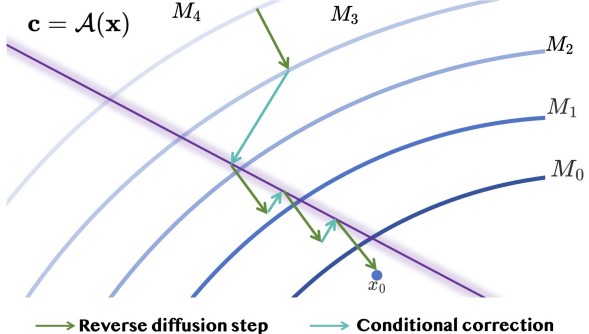

Figure 10: An illustration of the sampling process with conditional correction operator for inverse problem. The conditional correction process (cyan arrow) projects $\mathbf{x}_t$ onto the data manifold satisfying the constraint $\mathbf{c} = \mathcal{A}(\mathbf{x})$.

in inpainting tasks can be expressed as: $\mathbf{c}_y = (1 - \mathbf{c}_m) \odot \mathbf{x}$. Pioneer diffusion work SDE (Song et al., 2021b) performs inpainting based on conditional correction by replacing the unmask region in denoising output $\mathbf{x}_{0|t}$ with its counterpart in reference image $\mathbf{c}_y$ to ensure the faithfulness to the content in unmasked region. Different from SDE (Song et al., 2021b), Repaint (Lugmayr et al., 2022) prefers to perform replacement correction on latent variable $\mathbf{x}_t$. Besides, Repaint rolls back the intermediate latent variable $\mathbf{x}_t$ to the previous time step and resamples it back to time step $t$ several times to eliminate the artifacts caused by conditional correction. The constraint in super-resolution task can be written as: $\mathbf{c} = \phi_N \mathbf{x}$, where **c** denotes the low-resolution image of **x** downsampled by degradation matrix $\phi_N$ with factor $N$. ILVR (Choi et al., 2021) performs conditional correction by substituting the low-frequency components in the latent variable with its counterpart noisy low-resolution image to ensure the consistency between degraded latent variable and its counterpart noisy reference low-resolution image. Conditional correction operators are also widely employed in image editing tasks to preserve the background not requiring editing (Couairon et al., 2023; Patashnik et al., 2023; Wang et al., 2023a; Lin et al., 2024b; Huang et al., 2023b).

Given a background mask in the source image, text-based image editing can be formulated as an image inpainting task, where the masked foreground region is synthesized based on the provided text prompt. However, such foreground or background masks are often unavailable in real-world editing scenarios. To address this, a line of work proposes automatic mask or segmentation generation by inferring plausible layouts for the user-desired edited image, conditioned on both the source image and the text prompt. DiffEdit (Couairon et al., 2023) identifies background masks by comparing the denoising outputs of a noisy source image when conditioned on the source prompt versus the target prompt. Object-Shape Variation (Patashnik et al., 2023) segments the source image by clustering attention maps into semantically coherent regions and aligning them with noun tokens in the prompt based on similarity with their corresponding cross-attention maps. Additionally, several approaches (Wang et al., 2023a; Lin et al., 2024b; Huang et al., 2023b) utilize pre-trained image segmentation models to automatically generate masks or segmentations, leveraging the structural information from both the source image and the text prompt.

CCDF (Chung et al., 2022b) proposes a general conditional correction formula for constraints in the form of a general noisy linear inverse problem. In practice, the conditional correction operator in (Song et al., 2021b; Lugmayr et al., 2022; Choi et al., 2021) can be expressed in the general form provided by CCDF. Besides, CCDF provides a theoretical basis for the faithfulness of this corrected sampling trajectory to original sampling process. CCDF proves when the linear degradation operator $\boldsymbol{H}$ is a non-expansive mapping, the upper bound of the deviation in final output $\mathbf{x}_0$ will converge to a constant as the total diffusion step $T \to \infty$. MCG (Chung et al., 2022a) further performs guidance on conditional correction framework provided by CCDF, which alleviates the deviation from original sampling process caused by conditional correction.

## 5 Challenges and Future Directions

Although DM-based conditional image synthesis has made remarkable progresses in generating high-quality images aligned with various user-provided conditions, there remains a significant disparity between academic advancements and practical needs for conditional image synthesis. In this section, we summarize several main challenges in this field and identify potential solutions to address them in the future.

### 5.1 Sampling Acceleration

The time-consuming sampling process often creates a bottleneck of diffusion-based image synthesis, and its acceleration will facilitate the model deployment in practice (Li et al., 2024c; Zhao et al., 2023b). Early works on sampling acceleration are devoted to reducing the number of sampling steps with better numerical solvers (Song et al., 2021a; Lu et al., 2022a;b; Zhou et al., 2024a; Chen et al., 2024a) or distilling pre-trained diffusion models to build short-cuts that enable faster sampling (Salimans & Ho, 2022; Meng et al., 2023; Song et al., 2023c; Zhou et al., 2024b). However, too few denoising steps in the distilled models may compromise the effectiveness of in-sampling condition integration. One feasible solution is to first train a model to approximate the conditional denoising outputs along the sampling process equipped with in-sampling conditioning mechanisms, and then perform distillation on this model (Meng et al., 2023). Another important line of existing works reduces the computational cost of each denoising step by decreasing model parameters using techniques such as knowledge distillation (Chen et al., 2021; 2022) and architecture search (Li et al., 2024c; Kim et al., 2023a; Zhao et al., 2023b). Most of DM-based parameter compression approaches are currently tailored for text-to-image models. Analyzing whether the parameter redundancy also exists for models of other conditional synthesis tasks, similar to those in text-to-image models, and extending these model compression methods to more complicated downstream tasks, is another promising future direction.

### 5.2 Artifacts Caused by In-sampling Conditioning Mechanisms

In-sampling conditioning mechanisms summarized in Sec. 4 allow for flexible condition integration in DM-based image synthesis without performing time-consuming condition integration for the denoising network. However, these conditioning mechanisms introduce modification to the standard sampling process in diffusion framework and lead to deviations from the modeled data distribution, which results in artifacts in synthesized images (Parmar et al., 2023; Lugmayr et al., 2022; Bansal et al., 2023; Yu et al., 2023). The vast majority of works resort to complex adjustment mechanisms to address the artifact issue caused by in-sampling condition integration. This includes time-step rolling back for guidance (Yu et al., 2023), localization for attention maps (Cao et al., 2023; Lu et al., 2023) and diffusion process revision for restoration tasks (Luo et al., 2023; Kawar et al., 2022). However, these methods are highly customized based on specific application scenarios. A feasible future direction for developing more generic solution is to perform lightweight fine-tuning on the denoising network with the diffusion loss based on the intermediate latent variables in the sampling process equipped with in-sampling conditioning mechanisms. This tends to smooth out artifacts under in-sampling conditioning mechanisms and synthesize desired images in a lower computational cost compared to performing condition integration in the denoising network.

### 5.3 Training Datasets

Among the various conditioning mechanisms, the most fundamental and effective pathway for condition integration is still the supervised learning on pairs of conditional input and image. Although training datasets are relatively sufficient for conditional synthesis tasks involving single-modality conditional inputs, such as text-to-image (Schuhmann et al., 2021; 2022), restoration (Agustsson & Timofte, 2017; Nah et al., 2017; Karras et al., 2019), and visual signal to image (Lin et al., 2014; Caesar et al., 2018; Zhou et al., 2017), gathering enough data for tasks with complex, multi-modal conditional inputs like image editing, customization, and composition remains challenging. With the advancement of training and efficient fine-tuning techniques for large language models, various types of large models are constantly being developed with powerful multi-modal representation learning (Brown et al., 2020; Li et al., 2022b; 2023b) and content generation abilities (Hertz et al., 2023; Tumanyan et al., 2023), making it possible to leverage these pre-trained models to automatically produce desired training datasets. We may also consider self-supervised or weakly supervised learning to reduce the demand for a large amount of high-quality training data (Zhang et al., 2023d; Xie et al., 2023b; Zhang et al., 2024e).

### 5.4 Robustness

Due to the lack of objective task-specific evaluation datasets and metrics in some complex tasks, studies for these tasks prefer to compare models based on a set of self-defined conditional inputs, making the performance appear overly optimistic. In fact, many renowned text-to-image models (Ramesh et al., 2022; Saharia et al., 2022b; Rombach et al., 2022) have been found to produce unsatisfactory synthesized results for certain specific categories of text prompts, as demonstrated by the shortcomings of Imagen (Saharia et al., 2022b) in generating facial images. Here we point out some pathways to address issues of robustness. First, for conditional inputs where the model performs poorly, augmenting the training dataset is a direct approach. Second, the difficulties in handling conditional inputs in a certain category may be due to the insufficient capability or unsuitability of the conditional encoder for this category of data. In this case, incorporating encoder architectures tailored for this data category into the conditional encoder, or designing more capable compound conditional encoders, becomes a preferable choice. Besides, performing specialization for given conditional inputs is also an effective pathway to provide robust results at the cost of time-consuming fine-tuning or optimization. Finally, sampling process conditioning mechanisms, such as guidance, conditional correction, and attention manipulation, can also be employed to achieve more detailed control and prevent undesired synthesis results.

### 5.5 Ethic considerations

The developments in AI-generated content (AIGC) propelled by the superior performance of diffusion-based conditional synthesis and their downstream applications lead to severe ethic considerations in aspects of bias and fairness, copyright, and the risk of exposure to harmful content. Safety-oriented DM-based conditional image synthesis is dedicated to mitigating these issues by embedding watermarks that are easily reproducible in DM-generated images to detect copyright infringement (Yuan et al., 2024; Cui et al., 2023; Wen et al., 2023), and reducing bias by increasing model's orientation towards minority groups in basic unconditional or text-conditioned synthesis via classic conditioning mechanisms, such as fine-tuning (Shen et al., 2023), guidance (Um et al., 2024), and conditional correction (Li et al., 2024a). Efforts have also been made in preventing harmful contents in the text-to-image task via harmful prompt detection (Rombach et al., 2022), prompt engineering (Li et al., 2024a) and safety guidance (Schramowski et al., 2023). The current safety-focused efforts mainly concentrate on basic unconditional or text-conditioned synthesis. We believe that for more complex conditional synthesis scenarios, safety-oriented efforts in this area can be focused on four main aspects: (a) detecting harmful conditional inputs, (b) filtering and removing bias from the training dataset, (c) providing safety-focused guidance for the sampling process, and (d) implementing safety-focused fine-tuning of the denoising network.

### 5.6 Unified conditional synthesis framework

While current research has developed diverse diffusion-based frameworks (Saharia et al., 2022c; Rombach et al., 2022; Hertz et al., 2023; Li et al., 2023i) specialized for specific conditional synthesis tasks, this often formalizes complex user intents into strict categories, which limits flexibility and ease of use amidst the rapidly growing number of conditional image synthesis models and tasks. A highly appealing future direction is developing a unified framework that allows users to specify tasks, conditional inputs, and desired outputs flexibly. The recent advent of powerful multimodal large language models (MLLMs) represented by GPT-4o[1] and Gemini 2.0 Flash[2] offers a promising path. Leveraging their strong language understanding and integrated image processing capabilities, MLLMs can interpret various user intents and potentially unify diverse synthesis tasks within a single framework.

However, current MLLMs still exhibit limitations in image generation quality and control, represented by detail inaccuracies in restoration and editing, and difficulties with complex scenes. While the closed nature of SOTA MLLMs hinders detailed analysis, we hypothesize these limitations might stem from processing input images by aligning them to a semantic space, potentially neglecting fine-grained local details. Based on the impressive ability of current state-of-the-art MLLMs to understand user intent and synthesize corresponding images, we believe there is an urgent need to develop a powerful open-source MLLM in this field. Additionally, integrating more local details in user-provided images during conversations into the image generation modules of large models is also a promising future direction.

## 6 Conclusion

This survey presents a thorough investigation of DM-based conditional image synthesis, focusing on framework-level construction and common design choices behind various conditional image synthesis problems across seven representative categories of tasks. Despite the progress made, efforts are still needed in the future to handle challenges in practical applications. Future research should focus on collecting and constructing high-quality, unbiased, and task-specific datasets, as well as designing effective conditional encoder architectures and in-sampling conditioning mechanisms to enable robust and accurate conditional modeling for generating stable and high-fidelity results. The trade-off between fast sampling and synthesis quality also remains a critical challenge for real-world deployment. In addition, exploring unified conditional synthesis frameworks built upon state-of-the-art Multimodal Large Language Models (MLLMs) offers a promising direction, as these models can seamlessly integrate diverse control signals and provide generalizable conditioning across a wide range of tasks. Finally, as a popular AIGC technology, it is necessary to fully consider the safety issues and legitimacy it brings.

## Acknowledgement

Zheyuan Zhan and Can Wang are supported by the National Natural Science Foundation of China (No. 62476244), ZJU-China Unicom Digital Security Joint Laboratory and the advanced computing resources provided by the Supercomputing Center of Hangzhou City University. Jian-Ping Mei is supported by the National Natural Science Foundation of China (Grant No. 62276234), and Zhejiang Provincial Natural Science Foundation (Grant No: ZCLZ24F0202).

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

# A   Appendix

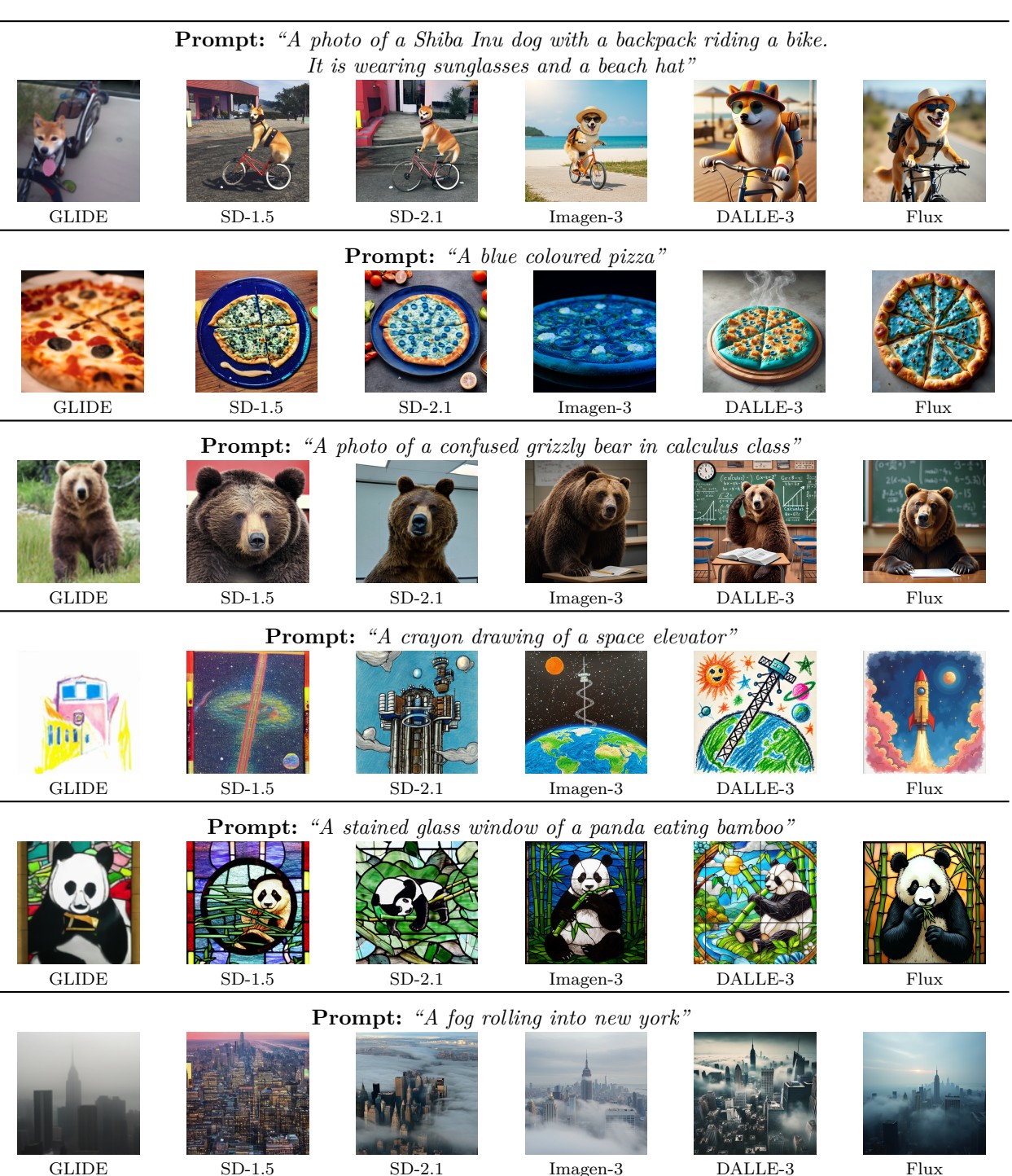

Figure 11: Visual comparison of the classic diffusion-based works for text-to-image, including GLIDE (Nichol et al., 2022), Stable Diffusion (Rombach et al., 2022), Imagen (Ho et al., 2022a), DALLE (Ramesh et al., 2022), and Flux (Black-Forest, 2024).

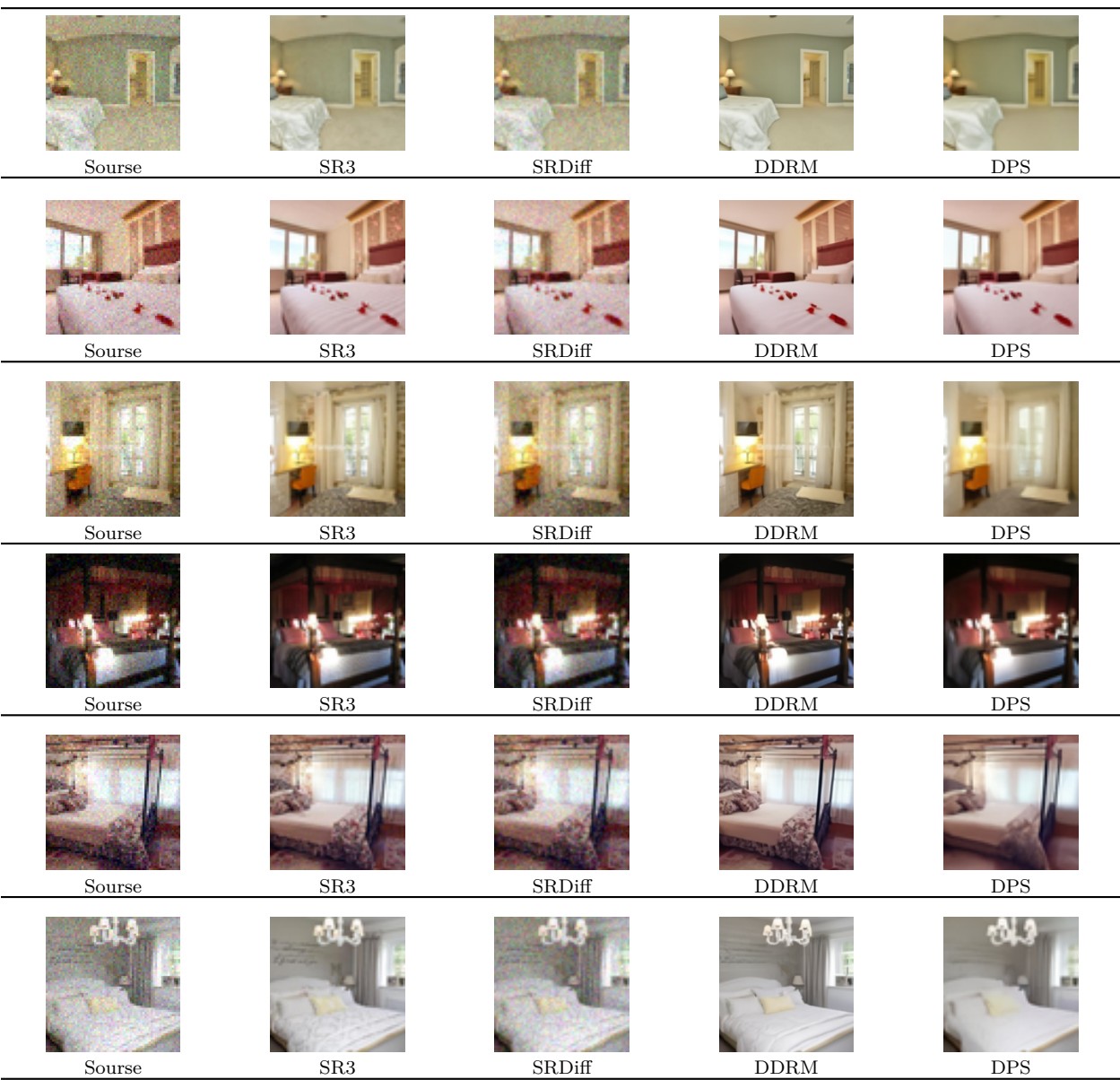

Figure 12: Visual comparison of the classic diffusion-based works for super-resolution restoration, including SR3 (Saharia et al., 2022c)), SRDiff (Li et al., 2022a), DDRM (Kawar et al., 2022), and DPS (Chung et al., 2023b).

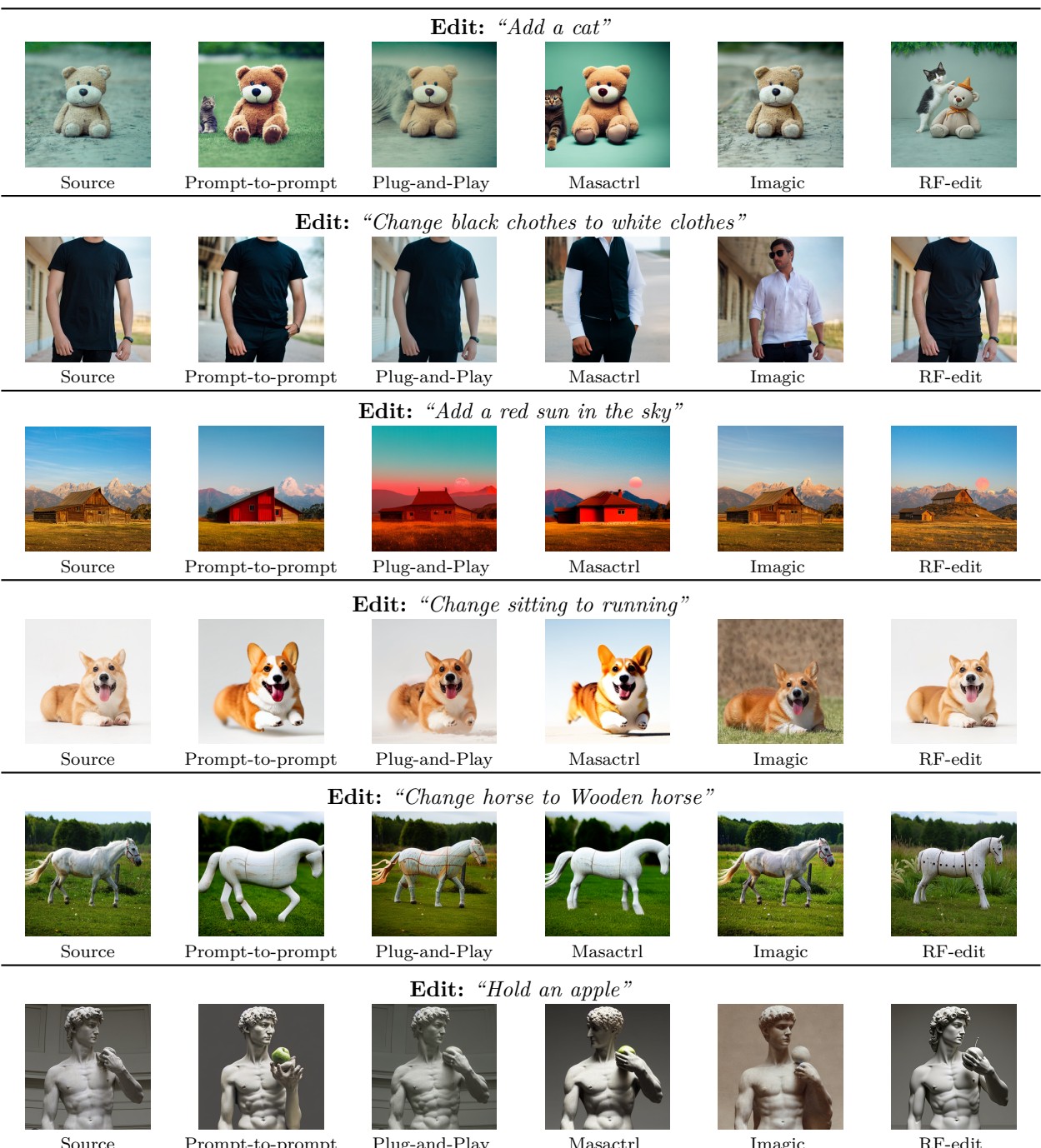

Figure 13: Visual comparison of the classic diffusion-based works for image editing, including Prompt-to-prompt (Hertz et al., 2023), Plug-and-play (Tumanyan et al., 2023), Masactrl (Cao et al., 2023), Imagi (Kawar et al., 2023), and RF-edit (Wang et al., 2024a).

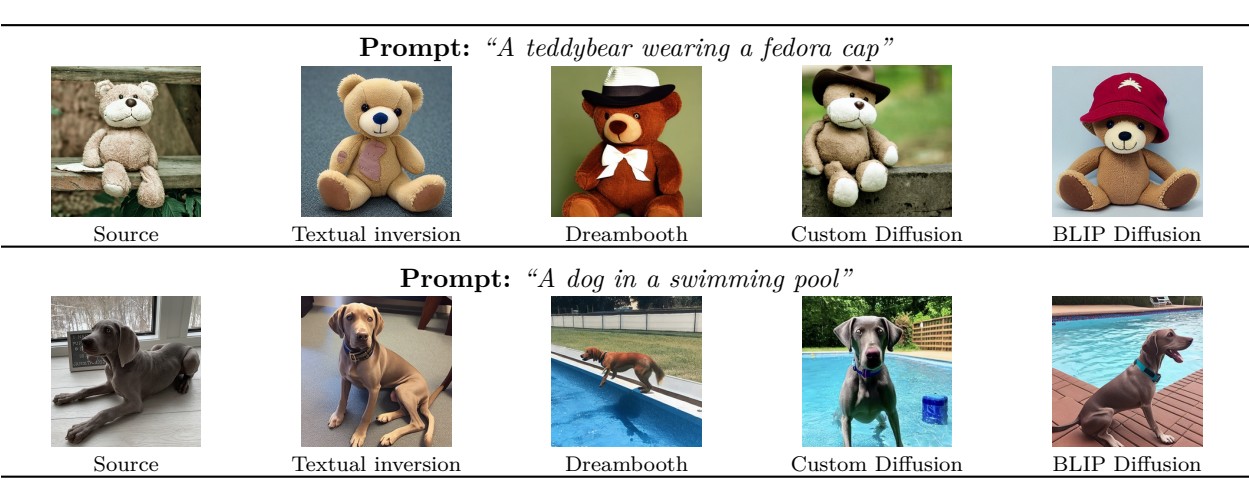

**Prompt:** *"A cute cat, high quality, extremely detailed"*

| Source | Visual Signal | Readout Guidance | T2I-Adapter | X-adapter | ControlNet |

**Prompt:** *"Flower,best quality, extremely datailed"*

| Source | Visual Signal | Readout Guidance | T2I-Adapter | X-adapter | ControlNet |

**Prompt:** *"Professional headshot of a woman with the eiffel tower in the background"*

| Source | Visual Signal | Readout Guidance | T2I-Adapter | X-adapter | ControlNet |

**Prompt:** *"A beautiful living room"*

| Source | Visual Signal | Readout Guidance | T2I-Adapter | X-adapter | ControlNet |

Figure 14: Visual comparison of the classic diffusion-based works for depth to image (visual signal-to-image), including Readout Guidance (Luo et al., 2024), T2I-Adapter (Mou et al., 2024c), X-adapter (Ran et al., 2024), and ControlNet(Zhang et al., 2023b).

**Prompt:** *"A teddybear wearing a fedora cap"*

| Source | Textual inversion | Dreambooth | Custom Diffusion | BLIP Diffusion |

**Prompt:** *"A dog in a swimming pool"*

| Source | Textual inversion | Dreambooth | Custom Diffusion | BLIP Diffusion |

Figure 15: Visual comparison of the classic diffusion-based works for Customization, including Textual Inversion (Gal et al., 2023a), DreamBooth (Ruiz et al., 2023), Custom Diffusion (Kumari et al., 2023), and BLIP Diffusion (Li et al., 2023a).

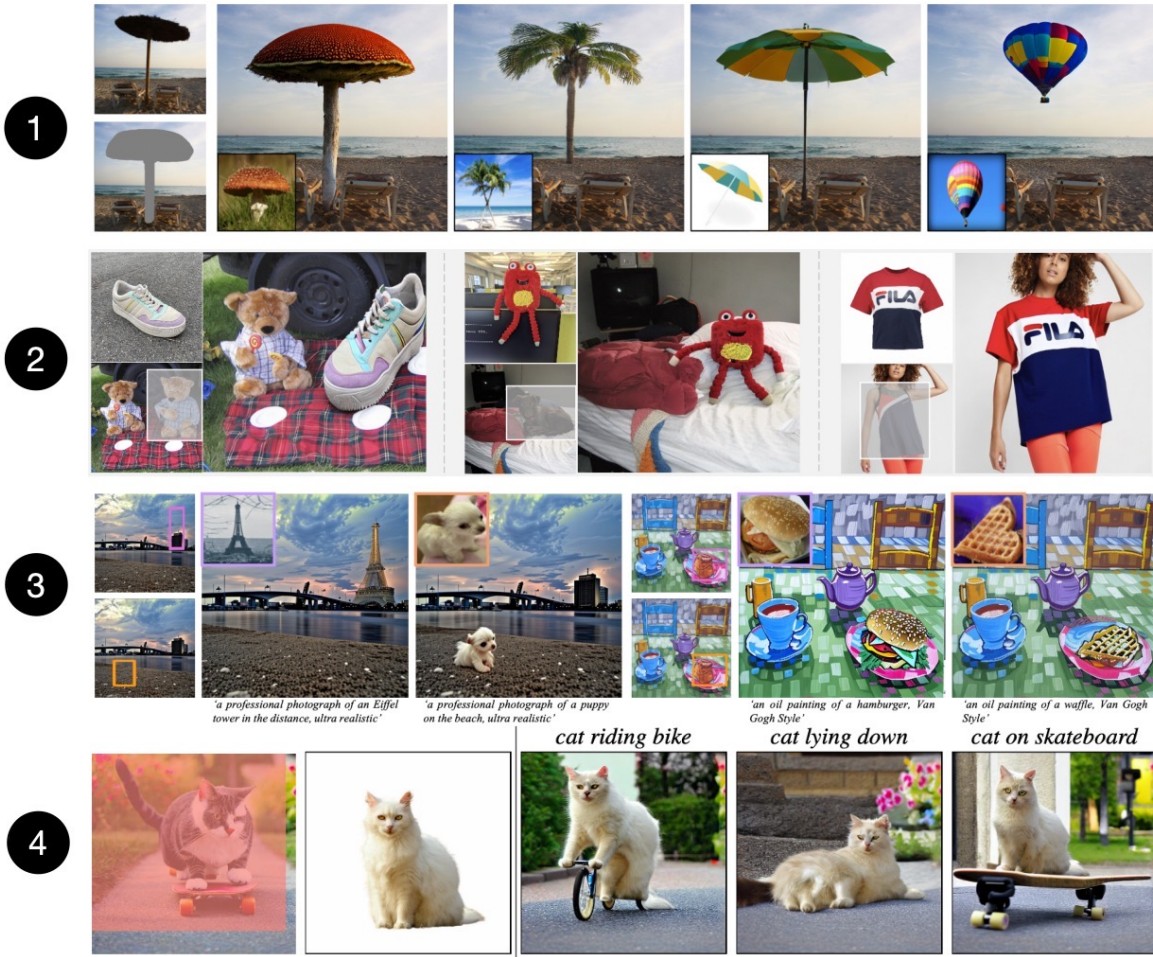

Figure 16: Visual comparison of the classic diffusion-based works for Customization, including 1. PbE (Zhang et al., 2023d), 2. DreamInpainter (Xie et al., 2023b), 3. TF-ICON (Lu et al., 2023), 4. Anydoor (Chen et al., 2024c).

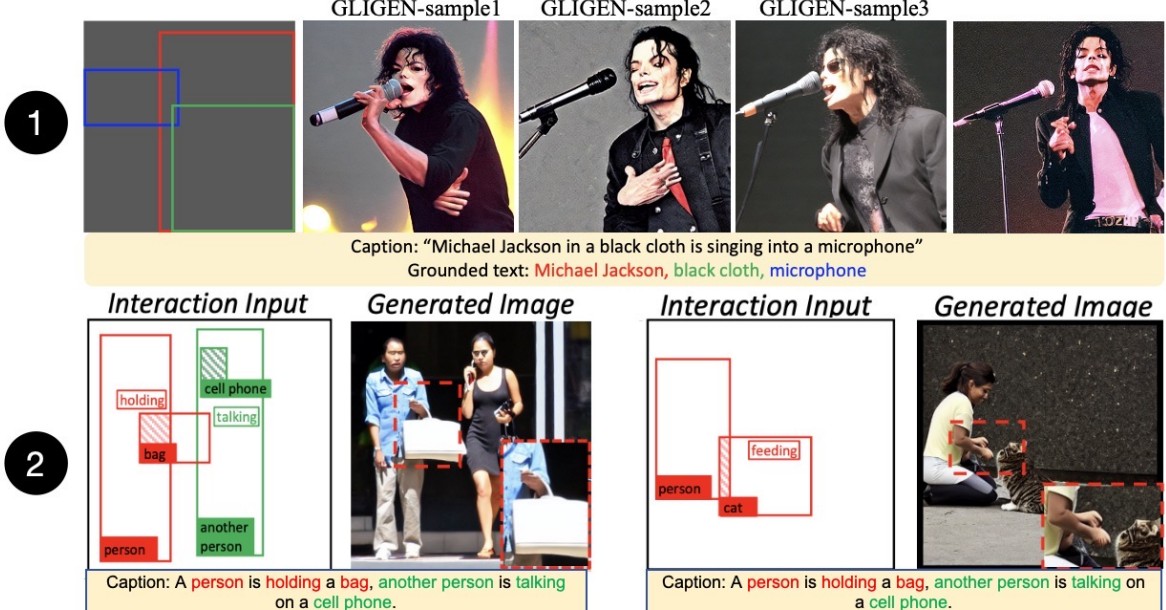

Figure 17: Visual comparison of the classic diffusion-based works for Customization, including 1. GLIGEN (Li et al., 2023i), 2. InteractDiffusion (Hoe et al., 2023).

Table 4: The corresponding works for the various stacks of conditioning mechanisms shown in Tab.1.

| Stack of conditioning mechanisms for denoising network | |
|---|---|
| **Serial Number** | **Model** |
| **DN1** | Rombach et al. (2022); Saharia et al. (2022b); Nichol et al. (2022) Ramesh et al. (2022); Gu et al. (2022); Balaji et al. (2022) |
| **DN2** | Saharia et al. (2022c); Ho et al. (2022b); Li et al. (2022a) |
| **DN3** | Shang et al. (2024); Zhao et al. (2024a); Jiang et al. (2023a) Zheng et al. (2024a;b); Zhang et al. (2024d); Xue et al. (2024) |
| **DN4** | Fu et al. (2023); Huang et al. (2023c); Feng et al. (2023); Li et al. (2023e) |
| **DN5** | Brooks et al. (2023); Zhang et al. (2024c); Yildirim et al. (2023); Zhang et al. (2024b) Geng et al. (2023); Sheynin et al. (2024); Li et al. (2023f) |
| **DN6** | Kawar et al. (2023); Wu et al. (2023); Zhang et al. (2023c); Mou et al. (2024b) Mahajan et al. (2024); Ravi et al. (2023); Bodur et al. (2024); Li et al. (2023c); Zhang et al. (2023f) |
| **DN7** | Xiao et al. (2023); Ma et al. (2024); Shi et al. (2024a); Gal et al. (2023b) Jia et al. (2023); Lu et al. (2024); Shiohara & Yamasaki (2024) |
| **DN8** | Ruiz et al. (2023); Gal et al. (2023a); Kumari et al. (2023) Gu et al. (2024); Liu et al. (2023d;e); Han et al. (2023) |
| **DN9** | Mou et al. (2024c); Zhang et al. (2023b;d); Goel et al. (2023) Qin et al. (2023); Yang et al. (2023b); Zhao et al. (2024b); Ran et al. (2024); Jiang et al. (2023b) |
| **DN10** | Wang et al. (2022a); Xu et al. (2024) |
| **DN11** | Li et al. (2023h); Zeng et al. (2024); Kim et al. (2024) |
| **DN12** | Yang et al. (2023a); Xie et al. (2023a); Wang et al. (2023b); Xie et al. (2023b) Song et al. (2023d); Kim et al. (2023b); Chen et al. (2024c) |
| **DN13** | Li et al. (2023i); Hoe et al. (2023); Wang et al. (2024b) |
| Stack of conditioning mechanisms for sampling process | |
| **Serial Number** | **Model** |
| **SP1** | Tian et al. (2024); Liu et al. (2023b) |
| **SP2** | Luo et al. (2023); Yue et al. (2024); Welker et al. (2024); Wang et al. (2024d); Delbracio & Milanfar (2023) |
| **SP3** | Kawar et al. (2021; 2022); Wang et al. (2024c) |
| **SP4** | Avrahami et al. (2022); Chung et al. (2022a); Song et al. (2023a) Chung et al. (2023a); Fei et al. (2023); Rout et al. (2024b) |
| **SP5** | Lugmayr et al. (2022); Choi et al. (2021); Chung et al. (2022b) |
| **SP6** | Su et al. (2023); Meng et al. (2022a); Mokady et al. (2023); Dong et al. (2023); Wang et al. (2023d); Wallace et al. (2023); Miyake et al. (2023); Ju et al. (2023); Meiri et al. (2023); Brack et al. (2024); Wu & De la Torre (2023); Huberman-Spiegelglas et al. (2023); Nie et al. (2023); Zhang et al. (2023e) |
| **SP7** | Couairon et al. (2023); Yang et al. (2023c); Patashnik et al. (2023) Wang et al. (2023a); Lee et al. (2024); Huang et al. (2023b) |
| **SP8** | Hertz et al. (2023); Tumanyan et al. (2023); Cao et al. (2023) Lu et al. (2023); Chung et al. (2024); Guo & Lin (2023) |
| **SP9** | Parmar et al. (2023); Mou et al. (2024b); Lin et al. (2023); Park et al. (2024) |
| **SP10** | Voynov et al. (2023); Singh et al. (2023); Luo et al. (2024); Mo et al. (2024) |
| **SP11** | Zhao et al. (2023a); Shirakawa & Uchida (2024); Bar-Tal et al. (2023) |
| **SP12** | Cao et al. (2023); Patashnik et al. (2023); Lu et al. (2023) Balaji et al. (2022); Liu et al. (2023e); Guo & Lin (2023) |
| **SP13** | Chen et al. (2024b); Epstein et al. (2024); Mou et al. (2024a) |
| **SP14** | Liu et al. (2022) |
| **SP15** | Ho & Salimans (2022); Sadat et al. (2024); Kynkäänniemi et al. (2024) |
| **SP16** | Yu et al. (2023); Bansal et al. (2023) |

