# OpenReview forum: "Conditional Image Synthesis with Diffusion Models: A Survey"
_TMLR — Accepted by TMLR_

### Review · Reviewer_UXiJ · 2024-12-26

**Summary Of Contributions:**

This is a survey paper encompassing early and recent works on conditional image generation with diffusion models. The survey categorizes, organizes the taxonomy and overviews the different conditioning mechanisms, with two main branches– the network itself and the sampling process. The survey also covers the various applications of these conditioning mechanisms and highlights open questions for future research.

**Audience:**

Yes

**Claims And Evidence:**

Yes

**Requested Changes:**

* Please proofread and fix all the grammatical errors in the text. I pointed out some of them under “Minor”. You can try pasting the text in Word or Google Docs to quickly locate obvious mistakes, but some mistakes require actually reading the paper.
* See “Weaknesses” and “Minor” above.

**Strengths And Weaknesses:**

**Strengths**:
* Overall, the motivation for this survey is clear and it presents an organized taxonomy while distinguishing itself from previous surveys.
* Covers many recent works which could serve as reference for future work on conditional image synthesis.

**Weaknesses**:
* In general, diffusion models work with various deformations, not only Gaussian noise ([1], [2]) as mentioned throughout the manuscript (e.g., page 1 - second paragraph, Section 2 - first paragraph). It is worth noting that while most diffusion models indeed utilize Gaussian noise, it is possible to learn a mapping from *some* distribution to the real data distribution with a diffusion process.
* Figure 2 is not entirely clear – I understand what the authors try to convey, but I don’t think the figure is successful in doing that. Perhaps more explanation/legend on the colors of the arrows and what they mean. I believe this figure needs to be reworked.
* Something is weird with the citation format, e.g., in Figure 1 (there is only a space between the method’s name and the authors) and page 2 - second paragraph. I think `\citep’ is not defined or something like that. Another example is the beginning of Section 2.1.2. Page 4 - “or cross-attention layersRombach et al. (2022)”.
* Writing could be improved (proofreading, see “Minor”). It really feels like the authors didn’t perform a final pass over the writing after the initial version.
* In page 8, you mention “and subsequently enlarge the resolution of the synthesized results.”. I believe you can  spare a few words on how this resolution enlargement happens (e.g., super-resolution pre-trained networks).
* I think the survey is a bit outdated, architecture-wise, as many modern models shifted to Diffusion Transformers (DiTs) and Adaptive Layer Normalization (AdaLN) conditioning.
* Section 4.1.1 writing: I found this section particularly hard to read; repeated usage of the word “However” (even to start sentences), arbitrary claims such as “which always leads to unsatisfied results especially under classifier-free guidance.” (“always”? I think it is a bit strong), and repeated usage of the term “classifier-free guidance”, which is common for those familiar with the literature, but not defined up to this point (only described later in Section 4.3.2). The differences between deterministic and stochastic inversion are not clear.
* Section 4.2 requires proofreading.
* Section 4.3.2 refers the reader to Section 4.5, which comes later in the paper, to understand “traditional guidance”. Given my comment above regarding guidance, perhaps a reordering of the sections is due.

[1] Bansal, Arpit, et al. "Cold diffusion: Inverting arbitrary image transforms without noise." Advances in Neural Information Processing Systems 36 (2024). (https://arxiv.org/abs/2208.09392)

[2] Heitz, Eric, Laurent Belcour, and Thomas Chambon. "Iterative α-(de) blending: A minimalist deterministic diffusion model." ACM SIGGRAPH 2023 Conference Proceedings. 2023. (https://arxiv.org/abs/2305.03486)

[3] Peebles, William, and Saining Xie. "Scalable diffusion models with transformers." Proceedings of the IEEE/CVF International Conference on Computer Vision. 2023. (https://arxiv.org/abs/2212.09748 )

**Minor**:
* Page 4, first paragraph, just before Section 2.1.2, for $\sigma_t$, is the relationship between the two square-roots multiplication?
* Since a large portion of the text is focused on the U-Net architecture, and this is an introductory survey, perhaps it will benefit from a figure of this architecture.
* Page 7, “The main challenge in modeling a effective text-to-image” -> “an effective”, “in (a) precisely capture the users’ intention described in text prompts and (b) build “ -> “capturing”, “building”. “in acceptable computational cost” -> “at acceptable computational costs”. “base on Transformer encoder” -> “based”.
* Page 10, “ as tje encoder for visual signals,” -> “the”.
* Page 11, “ some works design developed task-specific attention modules for condition injection” -> choose between “designed” and “developed”.
* Page 11, what is “a UniFusion block”?
* Page 12, “However, for conditional inputs contain multimodal components or intricate semantics” -> “inputs that contain”.
* Page 12, “Base on the fine-tuning strategy,” -> “Based”.
* Page 19, “ subsequently composing these noise to” -> “noises”.
* Page 21, “aligned with the definiation of” -> “definition”
* Page 22, “which are incorporate into “ -> “incorporated”, “and perform” -> “performing”
* Page 23, fix the sentence: “In practice, a branch of works employ guidance to control the consistency between diffuse output and given visual signal” (“employs”, “diffused output/diffusion outputs” [what is “diffuse output”?], “a/the given visual signal”). “diverse visual signal” -> “signals”.
* Page 24, “attention map as guide reference” -> “as a guidance”, “variable will be pass” -> “passed”.
* Page 25, “The constrain “ -> “constraint”

---

> ### Author Response · Authors · 2025-04-04
> **Response to Reviewer UXiJ (Part 1)**
>
> Thank you for your thorough and thoughtful review and insightful feedback. We appreciate your positive comments on the paper's organization, taxonomy, and contribution. We have made point-by-point revisions to address your concerns, as detailed below:
>
> >  **Requested Change 1: In general, diffusion models work with various deformations, not only Gaussian noise ([1], [2]) as mentioned throughout the manuscript (e.g., page 1 - second paragraph, Section 2 - first paragraph). It is worth noting that while most diffusion models indeed utilize Gaussian noise, it is possible to learn a mapping from *some* distribution to the real data distribution with a diffusion process.**
>
> Thank you for your insightful comments. We have added a discussion of this point at the beginning of Section 2 (In page 3), with the changes highlighted in red.
>
> >  **Requested Change 2: Figure 2 is not entirely clear – I understand what the authors try to convey, but I don’t think the figure is successful in doing that. Perhaps more explanation/legend on the colors of the arrows and what they mean. I believe this figure needs to be reworked.**
>
> Thank you for your insightful feedback. Based on your suggestions, we have updated this figure (now Figure 3 in the page 8). To enhance clarify, we have added an illustration at the bottom of the figure to indicate the meaning of each component.

---

> ### Author Response · Authors · 2025-04-04
> **Response to Reviewer UXiJ (Part 2)**
>
> > **Requested Change 3: Something is weird with the citation format, e.g., in Figure 1 (there is only a space between the method’s name and the authors) and page 2 - second paragraph. I think `\citep’ is not defined or something like that. Another example is the beginning of Section 2.1.2. Page 4 - “or cross-attention layersRombach et al. (2022)”.**
>
> Thank you for pointing out the issues with the citation format in our manuscript. We have updated all the citation formats accordingly.
>
> >**Requested Change 4: Writing could be improved (proofreading, see “Minor”). It really feels like the authors didn’t perform a final pass over the writing after the initial version.**
>
> Thank you very much for pointing out the shortcomings in our writing under the "Minor" section. We have thoroughly proofread the manuscript to improve its overall writing quality and have addressed each of the issues you mentioned. For details, please refer to our responses to the "Minor" section below.

---

> ### Author Response · Authors · 2025-04-04
> **Response to Reviewer UXiJ (Part 3)**
>
> > **Requested Change 5: In page 8, you mention “and subsequently enlarge the resolution of the synthesized results.”. I believe you can spare a few words on how this resolution enlargement happens (e.g., super-resolution pre-trained networks). .**
>
> Thank you for your insightful feedback. We have added relevant information at the end of the sentence regarding how this resolution enlargement happens (highlighted in red on page 10). In addition, a more detailed discussion on how text-to-image diffusion models perform upsampling can be found in the following sections where specific works are introduced.
>
> >**Requested Change 6: I think the survey is a bit outdated, architecture-wise, as many modern models shifted to Diffusion Transformers (DiTs) and Adaptive Layer Normalization (AdaLN) conditioning.**
>
> Thank you for your suggestion regarding the timeliness of our work. In the latest version, we have included the relevant studies you mentioned and introduced them in the last paragraph of Section 3.1.1 (highlighted in red on page 10).

---

> ### Author Response · Authors · 2025-04-04
> **Response to Reviewer UXiJ (Part 4)**
>
> > **Requested Change 7: Section 4.1.1 writing: I found this section particularly hard to read; repeated usage of the word “However” (even to start sentences), arbitrary claims such as “which always leads to unsatisfied results especially under classifier-free guidance.” (“always”? I think it is a bit strong), and repeated usage of the term “classifier-free guidance”, which is common for those familiar with the literature, but not defined up to this point (only described later in Section 4.3.2). The differences between deterministic and stochastic inversion are not clear.**
>
> Thank you for your feedback regarding the readability of Section 4.1.1. Based on your suggestions, we have reorganized Section 4.1.1. The revisions include:
>
> - Adjustments to the use of words such as "However" and "always" to improve the accuracy and scientific rigor.
> - Considering that classifier/classifier-free guidance is widely used in diffusion-based conditional image synthesis frameworks, we have added Section 2.4 in Section Background (highlighted in red on page 7) to discuss these methods in detail.
> - We have reorganized the discussion on different inversion methods. In particular, the differences between deterministic and stochastic inversion are now discussed in the part related to stochastic inversion (highlighted in red on page 19).
>
> >**Requested Change 8: Section 4.2 requires proofreading.**
>
> Thank you for your suggestion. We have reorganized the text in Section 4.2 and added recent works on attention manipulation-based image editing using Diffusion Transformers (DiTs) and Adaptive Layer Normalization (AdaLN) (highlighted in red on page 21 and 22).
>
> > **Requested Change 9: Section 4.3.2 refers the reader to Section 4.5, which comes later in the paper, to understand “traditional guidance”. Given my comment above regarding guidance, perhaps a reordering of the sections is due.**
>
> Thank you for your feedback regarding the organization of the paper. As mentioned in our response to Section 4.1.1, we have moved the discussion on classifier/classifier-free guidance earlier to Section 2.4 (highlighted in red on page 7) .

---

> ### Author Response · Authors · 2025-04-04
> **Response to Reviewer UXiJ (Part 5 for minors):**
>
> > **Requested Change 10: Page 4, first paragraph, just before Section 2.1.2, for σt, is the relationship between the two square-roots multiplication?**
>
> Thank you for pointing out the potential ambiguity in our notation. We have added a multiplication sign at this point to clarify that the relationship between the two square roots is multiplication (highlighted in red on page 4).
>
> > **Requested Change 11: Since a large portion of the text is focused on the U-Net architecture, and this is an introductory survey, perhaps it will benefit from a figure of this architecture. .**
>
> Thank you for your suggestion. We have added the figure of the U-Net architecture (Figure 2 in the lateset submission).
>
> >**Requested Change 12: The main challenge in modeling a effective text-to-image” -> “an effective”, “in (a) precisely capture the users’ intention described in text prompts and (b) build “ -> “capturing”, “building”. “in acceptable computational cost” -> “at acceptable computational costs”. “base on Transformer encoder” -> “based?**
>
> Thank you for pointing out the writing error. We have made the correction.
>
> > **Requested Change 13: as tje encoder for visual signals,” -> “the**
>
> Thank you for pointing out the spelling error. We have made the correction.
>
> > **Requested Change 14: some works design developed task-specific attention modules for condition injection” -> choose between “designed” and “developed**
>
> Thank you for pointing out the writing error. We have made the correction.
>
> > **Requested Change 15: what is “a UniFusion block**
>
> Thank you for your suggestion. In the latest version, we have provided a detailed description of the Unifusion block's structure (highlighted in red on the page 14).
>
> > **Requested Change 16: However, for conditional inputs contain multimodal components or intricate semantics” -> “inputs that contain**
> >
> > **Requested Change 17: Page 12, “Base on the fine-tuning strategy,” -> “Based”.**
> >
> > **Requested Change 18: Page 19, “ subsequently composing these noise to” -> “noises”.**
> >
> > **Requested Change 19: Page 21, “aligned with the definiation of” -> “definition”**.
> >
> > **Requested Change 20: Page 22, “which are incorporate into “ -> “incorporated”, “and perform” -> “performing”**
> >
> > **Requested Change 21: Page 23, fix the sentence: “In practice, a branch of works employ guidance to control the consistency between diffuse output and given visual signal” (“employs”, “diffused output/diffusion outputs” [what is “diffuse output”?], “a/the given visual signal”). “diverse visual signal” -> “signals”.**
> >
> > **Requested Change 22: Page 24, “attention map as guide reference” -> “as a guidance”, “variable will be pass” -> “passed”.**
> >
> > **Requested Change 23: Page 25, “The constrain “ -> “constraint”**.
>
> Thank you for your careful reading and for pointing out the writing errors. We have thoroughly proofread the entire paper and corrected all the errors you mentioned.

---

> > ### Comment · Reviewer_UXiJ · 2025-04-10
> > **Thank you for your effort**
> >
> > I thank the authors for their effort and for fixing all the issues. Overall, even though the survey is a bit outdated (but the field is moving very fast, so it is reasonable), I think there is a contribution to the community and I'm leaning towards accepting it.

---

### Review · Reviewer_U34P · 2025-01-16

**Summary Of Contributions:**

This paper presents a comprehensive survey of diffusion models for conditional image synthesis. After giving a short background on diffusion models, the paper provides a taxonomy of techniques based on the employed conditioning mechanisms, breaking it down into two fundamental categories:

1. **Integration in the network:** Methods that incorporate conditioning into the denoising network during training, re-purposing, or specialization.
2. **Integration in the sampling:** Methods that incorporate conditioning into the sampling process, including inversion, guidance, or attention manipulation.

Representative approaches are described in more detail, allowing to categorize and compare the rapidly increasing number of methods that have been developed in recent years. The paper concludes by highlighting open challenges and future directions in the field.

**Audience:**

Yes

**Broader Impact Concerns:**

One could add a paragraph on ethical considerations (e.g., bias mitigation strategies, safeguards against misuse, and potential risks of AI-generated content).

**Claims And Evidence:**

Yes

**Requested Changes:**

As mentioned in “weaknesses” above, my main suggestion would be to add further comparisons (beyond the taxonomy) to help the reader navigate the existing literature.

**Minor:**

- For most citations, `\citep` should be used instead of `\citet`
- `T2I` should be defined on p. 2
- It should be made clear that $\epsilon_\theta$ is defined to be the “noise-prediction” network and not the “denoising network” (i.e., predicting $x_0$) as mentioned on p. 4.

**Typos:**

- “ as describes in Sec. 3.2.2” → described
- “as tje encoder”
- "sampling process iteratively reserve noisy latent variable into desired image”
- “Noise-ddding inversion”
- “error accumulation .” (whitespace)

**Strengths And Weaknesses:**

### Strengths

1. **Taxonomy:** The paper is generally well-structured, covers a wide range of relevant papers, and categorizes them into clear taxonomies, which allows the reader to quickly navigate the growing literature. Moreover, the figures effectively illustrate different conditioning mechanisms.
2. **Applications:** Different from existing surveys, the material goes beyond text-to-image generation and covers a wide range of image synthesis tasks such as image restoration, image composition, and layout control.
3. **Future directions:** The survey provides an insightful outlook on open challenges and future directions in the field of conditional image synthesis with diffusion models.

### Weaknesses

1. **Comparisons:** The paper provides a great overview and *taxonomy* of methods. However, for a survey, it would also be good to provide further *comparisons*:
    - High-level (e.g., tabular) comparison of different approaches to understand their characteristics, e.g., advantages & drawbacks, applicability, computational costs, guarantees, etc.
    - While the main scope of the paper is conditional image synthesis, it would still be interesting to categorize which methods are tailored to image domains and which ones are more generally applicable.
    - Comparison of performance and effectiveness for different (real-world) applications. While I think that an empirical evaluation is out of scope for the current paper, one could still provide a more qualitative comparison.
2. **Related works:** Please find below some (recent) works which could be added and discussed:
    - TFG: Unified Training-Free Guidance for Diffusion Models (https://arxiv.org/pdf/2409.15761)
    - A Survey on Diffusion Models for Inverse Problems (https://arxiv.org/pdf/2410.00083)
    - Extending RePaint, it would be interesting to mention further resampling-based methods; see, e.g., https://arxiv.org/abs/2305.08995, https://arxiv.org/abs/2307.08123, https://arxiv.org/abs/2403.06054, and https://arxiv.org/abs/2407.01521.

---

> ### Author Response · Authors · 2025-04-04
> **Response to Reviewer U34P (Part1)**
>
> Thank you for your thorough and thoughtful review and insightful feedback. We appreciate your positive comments on the paper's taxonomy, applications and future directions. We have made point-by-point revisions to address your concerns, as detailed below:
>
> > **Requested Change 1: The paper provides a great overview and taxonomy of methods. However, for a survey, it would also be good to provide further comparisons**
>
> Thank you for your insightful feedback. Based on your suggestion, we conducted a further comparison of the DCIS methods included in our paper, as detailed below.
>
> > **Requested Change 1.1: The paper provides a great overview and taxonomy of methods. However, for a survey, it would also be good to provide further comparisons**
>
> Thank you for your insightful feedback. We have added Table 2 and Table 3 to the paper to provide a high-level comparison of the subcategories of methods in Section 3 (condition integration in the denoising network) and Section 4 (condition integration in the sampling process). These tables highlight key aspects such as computational costs, theoretical guarantees, and applicable scopes.
>
> > **Requested Change 1.2: While the main scope of the paper is conditional image synthesis, it would still be interesting to categorize which methods are tailored to image domains and which ones are more generally applicable.**
>
> Thank you for your insightful feedback. As image synthesis is a fundamental task in computer vision, many more complex visual computing and synthesis tasks build upon its extensions. Therefore, the methods for image synthesis introduced in this paper can be readily extended to more complex visual tasks, such as video synthesis, 3D scene generation, and motion generation. In our latest submission, we have explicitly pointed this out in the paper and cited relevant diffusion-based conditional synthesis works for more complex visual tasks (highlighted in blue on page 2 and 3).
>
> > **Requested Change 1.3: Comparison of performance and effectiveness for different (real-world) applications. While I think that an empirical evaluation is out of scope for the current paper, one could still provide a more qualitative comparison.**
>
> Thank you for your insightful feedback. We have added a comparison of representative DCIS works in the appendix, covering five major tasks: text-to-image, image restoration, visual signal-to-image, image editing, and customization. For image composition and layout control, due to the varying formats of conditional inputs across different works, a direct comparison is not feasible. Instead, we present the representative outputs from several classic DCIS works for these two tasks .

---

> > ### Author Response · Authors · 2025-04-04
> > **Response to Reviewer U34P (Part3 for minors)**
> >
> > > **Requested change 3: For most citations, `\citep` should be used instead of `\citet`**
> >
> > Thank you for pointing out the citation format issues in our manuscript. We have revised and updated all citation formats accordingly.
> >
> > > **Requested change 4: T2I should be defined on p. 2**
> >
> > Thank you for your suggestion. In the latest version of our submission (highlightedin blue on page 2).
> >
> > > **Requested change 5: It should be made clear that $\epsilon_{\theta}$ is defined to be the “noise-prediction” network and not the “denoising network” (i.e., predicting $x_0$) as mentioned on p. 4.**
> >
> > Thank you for your feedback. As we understand it, in diffusion models, predicting the added noise $\epsilon$ and predicting the clean image $x_0$ are essentially equivalent (in fact, they are related by a linear transformation). Therefore, we did not distinguish between the two in this work. Throughout the paper, we consistently refer to the denoising network as the noise predictor within the diffusion-based synthesis framework.

---

> ### Author Response · Authors · 2025-04-04
> **Response to Reviewer U34P (Part2)**
>
> > **Requested change 2:**
> >
> > **Related works: Please find below some (recent) works which could be added and discussed**
> >
> >   - TFG: Unified Training-Free Guidance for Diffusion Models (https://arxiv.org/pdf/2409.15761)
> >
> >   - A Survey on Diffusion Models for Inverse Problems (https://arxiv.org/pdf/2410.00083)
> >   - Extending RePaint, it would be interesting to mention further resampling-based methods; see, e.g., https://arxiv.org/abs/2305.08995, https://arxiv.org/abs/2307.08123, https://arxiv.org/abs/2403.06054, and https://arxiv.org/abs/2407.01521.
>
> Thank you for your feedback. We have included all the papers you mentioned within the scope of this survey (highlighted in blue with the corresponding page numbers indicated in the table below).
>
> | Papar                                                        | Page |
> | ------------------------------------------------------------ | ---- |
> | TFG (https://arxiv.org/pdf/2409.15761)                       | 27   |
> | A Survey on Diffusion Models for Inverse Problems (https://arxiv.org/pdf/2410.00083) | 2    |
> | DiffPIR (https://arxiv.org/abs/2305.08995)                   | 27   |
> | Resample (https://arxiv.org/abs/2307.08123)                  | 25   |
> | Diffusion Purification for Image Restoration (https://arxiv.org/abs/2403.06054) | 20   |
> | DAPS (https://arxiv.org/abs/2407.01521)                      | 27   |
>
> ##

---

> ### Author Response · Authors · 2025-04-04
> **Response to Reviewer U34P (Part4 for typos)**
>
> > **Requested change 6: "as describes in Sec. 3.2.2” → described**
> >
> > **Requested change 7: “as tje encoder”**
> >
> > **Requested change 8: "sampling process iteratively reserve noisy latent variable into desired image”**
> >
> > **Requested change 9: “Noise-ddding inversion”**
> >
> > **Requested change 10:“error accumulation .” (whitespace)**
>
> Thank you for pointing out the writing issues in our paper. We have reviewed and corrected all the typos you mentioned.

---

> > ### Comment · Reviewer_U34P · 2025-04-10
> >
> > Thank you for your extensive revision and the response. I think that the additional tables, references, and comparisons significantly improve the contribution of the paper. While further methods and comparisons could be added, I acknowledge the large amount of literature in this area and I believe that the present version provides a helpful resource to the community.

---

### Review · Reviewer_RGBk · 2025-03-26

**Summary Of Contributions:**

This paper surveys the field of conditional image synthesis using diffusion models. The survey categorizes methods into those that condition using the denoising network and those that condition at sampling time. Within each category, the authors subcategorize methods based on training, repurposing, and specialization for denoising network and different condition integration mechanisms for sampling methods. Finally, the authors discuss challenges and future directions in the field including improving efficiency, robustness, and safety in conditional image generation.

**Audience:**

Yes

**Claims And Evidence:**

Yes

**Requested Changes:**

- It would be beneficial to include a high-level discussion of the pros/cons of the subcategories of methods.
- The citation format is inconsistent throughout the paper. Please use the commands \citet and \citep properly, e.g. "\citet{} proposes X" and "Recent works have studied conditional image synthesis using diffusion models \citep{}".

**Strengths And Weaknesses:**

## Strengths
- The survey is very comprehensive and covers a large number of methods categorized into denoising network-based or sampling-based.
- The taxonomy choice is clear and concise and describes well the different types of approaches for conditional image synthesis using diffusion models
- The challenges and future directions section is insightful and concretely presents several fruitful research directions.
- The paper is generally easy to follow and is well-written and organized.
- Table 1 is very well formatted and makes clear the conditional mechanisms within each category.
- Rather than just focusing on a subset of conditional models such as text-to-image diffusion, this survey covers general conditioning mechanisms.

## Weaknesses
- Although the categorization is done well, given the large number of methods discussed, there is no high-level discussion about the comparisons between the methods and the pros/cons of various methods.

---

> ### Author Response · Authors · 2025-04-04
> **Response to Reviewer RGBk**
>
> We truly appreciate your thorough review and your recognition of the strengths of our work. Please find our point-by-point responses to your comments below. We have made point-by-point revisions to address your concerns, as detailed below:
>
> > **Requested Change 1: It would be beneficial to include a high-level discussion of the pros/cons of the subcategories of methods.**
>
> Thank you for your insightful feedback. We have added Table 2 and Table 3 to the paper to provide a high-level comparison of the subcategories of methods in Section 3 (condition integration in the denoising network) and Section 4 (condition integration in the sampling process). These tables highlight key aspects such as computational costs, theoretical guarantees, and applicable scopes.
>
> > **Requested Change 2: The citation format is inconsistent throughout the paper. Please use the commands \citet and \citep properly, e.g. "\citet{} proposes X" and "Recent works have studied conditional image synthesis using diffusion models \citep{}".**
>
> Thank you for pointing out the issues with the citation format in our manuscript. We have revised and updated all citation formats accordingly.

---

> ### Comment · Reviewer_RGBk · 2025-04-22
>
> I thank the authors for their response to my concerns. Overall, I think this is a comprehensive survey that is well-organized and written. Therefore, I lean toward acceptance.

---

### Decision · Action_Editor_Q2oL · 2025-04-25

**Recommendation:** Accept as is

**Comment:**

The paper is a survey of conditional image synthesis with diffusion models. It introduces a taxonomy on such models. The reviewers agree that the paper is well-written, clearly organized and comprehensive.

The paper certifies as a Survey, as it surveys conditional image synthesis with diffusion models in a structured and comprehensive way. The authors should submit a camera ready version where changes are not indicated by color.

**Audience:**

The paper has a potentially huge audience in the field of image synthesis and diffusion models.

**Claims And Evidence:**

The paper claims to provide a survey of conditional image synthesis with diffusion models. This claim is met as the paper reviews such models in a clearly organized and comprehensive manner.